# Merge to Remember: Sharpness-Aware Isotropic Merging for Continual Learning

## Abstract

Continual learning with large pre-trained models offers significant potential for cross-task knowledge accumulation, but faces critical challenges such as catastrophic forgetting and parameter interference, especially when historical data is unavailable. Existing approaches typically rely on sequential fine-tuning or model merging strategies, yet often overlook the impact of loss landscape sharpness and dominant singular value directions, which leads to subspace misalignment and severe knowledge forgetting. In this paper, we propose the Sharpness-Aware Isotropic Merging (SAIM) framework, which introduces targeted optimizations in both the fine-tuning and merging stages to address these issues. Specifically, SAIM consists of two synergistic modules: (1) a Sharpness-Aware Block Coordinate Descent (SA-BCD) optimizer that guides the model toward flatter minima and selectively updates the most task-sensitive parameters, thereby mitigating parameter interference and enhancing robustness; (2) an adaptive isotropic merging algorithm that dynamically balances the singular value spectrum across tasks, effectively preventing the model from overemphasizing any single task direction, maintaining balanced knowledge representation, and improving subspace alignment. Extensive experiments on vision and language benchmarks demonstrate that SAIM achieves 5-10% higher accuracy than existing methods and maintains robust performance as the number of tasks increases. Ablation studies further validate the effectiveness of the SA-BCD fine-tuning strategy in promoting flat minima and reducing parameter interference, as well as its compatibility with various merging approaches.

## 1 Introduction

With the rapid development of deep learning, large pre-trained models have become the cornerstone of numerous artificial intelligence applications (Radford et al., 2021; Dosovitskiy et al., 2020). However, when these models are required to continually learn from new tasks, they often face two major challenges: catastrophic forgetting and parameter interference (Kirkpatrick et al., 2017; Lopez-Paz & Ranzato, 2017). Catastrophic forgetting refers to the loss of previous task capabilities when learning new tasks, while parameter interference manifests as conflicts between update directions of different tasks, leading to degraded overall performance. Traditional continual learning methods typically adopt sequential fine-tuning, which enables adaptation to new tasks but, due to its sequential nature, causes the model to be biased towards recent tasks and forget knowledge from earlier ones (Buzzega et al., 2020; Zenke et al., 2017). Additionally, existing strategies such as replay buffers, regularization, or parameter expansion often require storing historical data or introducing extra parameters, creating difficulties in balancing efficiency and effectiveness, and failing to adequately address data privacy and storage efficiency concerns (Rusu et al., 2016; Yoon et al., 2017; Farajtabar et al., 2020).

In recent years, model merging techniques have shown great potential in continual learning scenarios, particularly excelling in applications with data privacy protection and storage constraints (Ilharco et al., 2022; Wang et al., 2024; Li et al., 2023). Task arithmetic (Ilharco et al., 2022) constructs multi-task models, MagMAX (Marczak et al., 2024) reduces parameter interference via maximum magnitude selection, orthogonal projection-based continual merging (Tang et al., 2025) further minimizes task interference through sequential projections, while mixture-of-experts approaches (Qiu et al., 2024) leverage dynamic gating and low-rank adaptation for robust continual integration. How-

ever, when applied to continual learning scenarios, these methods face two main challenges: on one hand, they typically require maintaining the cumulative results of all historical task vectors, increasing storage overhead (Izmailov et al., 2018; Wortsman et al., 2022); on the other hand, these methods have limitations in addressing parameter update conflicts between different tasks, resulting in severe interference issues and representation space distortion. More importantly, the fine-tuning and merging processes are often treated as independent stages, overlooking their potential synergy, which leads to loss landscape incompatibility and subspace misalignment, ultimately affecting knowledge retention and transfer efficiency.

To address these challenges, we propose a novel continual learning method—Sharpness-Aware Isotropic Merging (SAIM)—which jointly optimizes the fine-tuning and merging processes to achieve efficient continual knowledge acquisition and retention. SAIM is based on two key technical innovations:

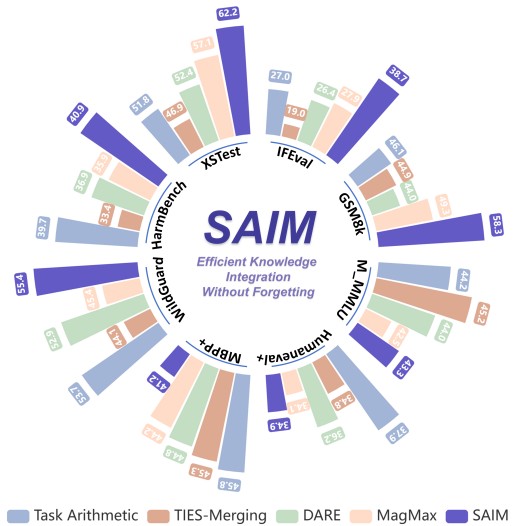

Figure 1: Performance comparison of SAIM against baseline methods on MergeBench using Llama-3.2-3B.

First, *in the fine-tuning stage*, we design a Sharpness-Aware Block Coordinate Descent (SA-BCD) optimizer, which guides the model towards flatter regions of the loss landscape and selectively updates only the top $p\%$ of parameters with the largest momentum. This approach explores the local geometry of the loss function to find parameter configurations that both adapt to new tasks and remain compatible with previous ones, laying a favorable foundation for subsequent merging. By seeking solutions with greater generalization ability in a broader solution space, the SA-BCD optimizer enhances the model's adaptability to different tasks, effectively mitigating the catastrophic forgetting problem.

Second, *in the merging stage*, we developed an adaptive isotropic merging algorithm specifically designed to address parameter interference between different tasks. This algorithm dynamically balances the singular value spectrum distribution, effectively preventing parameters from specific tasks from dominating in certain directions, ensuring fair representation of knowledge from all tasks in the merged model, thereby achieving higher subspace alignment across tasks. The algorithm dynamically adjusts the singular value balancing coefficient as tasks progress, efficiently integrating multi-task knowledge. Additionally, unlike traditional merging methods, SAIM does not require maintaining cumulative historical task vectors, which also simplifies storage requirements in continual learning scenarios. The effectiveness of our approach is illustrated in Figure 1, where SAIM consistently outperforms existing methods in both knowledge retention and adaptation in most tasks.

Our main contributions are as follows:

- **A joint fine-tuning and merging optimization framework:** We treat fine-tuning and merging as synergistic processes, where flat solutions facilitate subsequent parameter integration and merging provides a better starting point for new tasks. This unified framework enhances knowledge retention and improves overall continual learning performance.
- **Sharpness-Aware Isotropic Merging (SAIM) method:** We propose a novel approach that jointly optimizes fine-tuning and merging to mitigate catastrophic forgetting and parameter interference. By adaptively balancing the singular value spectrum, our method enhances subspace alignment across tasks, alleviates forgetting, and eliminates the need to store historical task vectors.
- **Extensive experimental validation:** We systematically evaluate SAIM across three vision architectures and three language model architectures, as well as varying numbers of tasks, demonstrating its effectiveness in reducing catastrophic forgetting and improving adaptation to new tasks. Furthermore, the SA-BCD can be combined with most existing model merging approaches to further enhance performance.

## 2 PRELIMINARIES

Before presenting our method, we briefly review the fundamental problem setup of continual learning and key theoretical foundations.

### 2.1 PROBLEM SETUP

In the continual learning (CL) setting, we focus on the following problem: Given a large pre-trained model with parameters $\theta_0$, and a sequence of $N$ downstream tasks $\mathcal{T} = \{1, 2, \ldots, N\}$ arriving sequentially, each task $t$ is associated with an independent dataset $D_t = \{(x_j^{(t)}, y_j^{(t)})\}_{j=1}^{n_t}$, where $x_j^{(t)} \in \mathcal{X}$ and $y_j^{(t)} \in \mathcal{C}_t \subset \mathcal{Y}$, with $\mathcal{C}_t$ being the label space for task $t$. Our goal is to efficiently integrate the newly obtained task expert $\theta_t$ with the current model parameters $\theta_{t-1}^{merged}$ at each step, and finally obtain a unified model compatible with all seen tasks.

As in conventional continual learning, we assume that no historical training data is accessible during the model integration process; all knowledge transfer and integration occur in parameter space. The continual Merging process can be formulated as:

$$\theta_t^{merged} = \text{ContinualMerge}(\theta_{t-1}^{merged}, \theta_0, \theta_t) \tag{1}$$

where $\theta_0^{merged} = \theta_0$. The final objective is to minimize the average loss over all seen tasks:

$$\min_{\theta_N^{merged}} \frac{1}{T} \sum_{t=1}^{T} \mathbb{E}_{(x,y) \sim \mathcal{D}_\sqcup} \mathcal{L}(h_t(\mathcal{F}(x; \theta_N^{merged})), y) \tag{2}$$

where $\mathcal{F}(x; \theta)$ denotes the backbone network, $h_t$ is the task-specific head, and $\mathcal{L}$ is the loss function.

### 2.2 SHARPNESS-AWARE MINIMIZATION AND FLATNESS

In the process of fine-tuning and merging deep models, the "flatness" of the local minima where model parameters reside is closely related to generalization. Flat minima are more robust to parameter perturbations, helping the model accommodate knowledge from multiple tasks and reducing catastrophic forgetting.

Sharpness-Aware Minimization (SAM, Foret et al., 2021) optimizes parameter flatness by simultaneously minimizing the loss value and its sharpness. The optimization objective is:

$$\min_{\theta} \left[ \max_{\|\epsilon\|_2 \leq \rho} \mathcal{L}(\theta + \epsilon; \mathcal{D}) - \mathcal{L}(\theta; \mathcal{D}) \right] + \mathcal{L}(\theta; \mathcal{D}) \tag{3}$$

where $\epsilon$ is a perturbation vector and $\rho$ controls the radius of the neighborhood. For efficiency, SAM uses a Taylor approximation to simplify the objective as:

$$\min_{\theta} \mathcal{L}(\theta + \hat{\epsilon}; \mathcal{D}), \quad \text{where} \quad \hat{\epsilon} = \rho \frac{\nabla_\theta \mathcal{L}(\theta; \mathcal{D})}{\|\nabla_\theta \mathcal{L}(\theta; \mathcal{D})\|} \tag{4}$$

By optimizing with SAM, the model is more likely to converge to flat minima, improving robustness and multi-task compatibility.

### 2.3 TASK ARITHMETIC AND SUBSPACE ALIGNMENT

Task Arithmetic (TA, Ilharco et al., 2022) provides a framework for constructing multi-task models. The knowledge of each task $t$ can be represented as a task vector $\tau_t = \theta_t - \theta_0$. By weighted summation of these task vectors, multiple task models can be merged:

$$\theta_{merged} = \theta_0 + \sum_{t=1}^{T} \alpha_t \tau_t \tag{5}$$

where $\alpha_t$ denotes the task weight. However, as the number of tasks increases, simple vector addition can lead to parameter conflicts and interference (Tang et al., 2025).

Recent studies have found that merging performance depends on the degree of alignment among the subspaces spanned by the task vectors (Gargiulo et al., 2025). The Subspace Alignment Ratio (SAR) measures the overlap between the merged model and task-specific parameter updates:

$$\text{SAR}(\Delta_{\text{src}}, \Delta_{\text{trg}}; k) = \frac{\|\pi_{k,\text{trg}}\Delta_{\text{src}}\|_F}{\|\Delta_{\text{src}}\|_F} \tag{6}$$

where $\pi_{k,\text{trg}}$ is the projection operator onto the principal subspace of the target update matrix. A higher SAR value indicates better alignment and typically improved merging performance.

## 3 APPROACH

Continual learning traditionally relies on sequential fine-tuning, using regularization (e.g., EWC (Kirkpatrick et al., 2017)) or experience replay (Buzzega et al., 2020) to mitigate catastrophic forgetting. Recently, model merging methods such as MagMAX (Marczak et al., 2024) and OPCM (Tang et al., 2025) have enabled continual learning by integrating knowledge from multiple task-specific models, thereby further reducing forgetting. However, most approaches treat fine-tuning and merging as separate processes. In this work, we explicitly consider both sharpness-aware fine-tuning (Sec. 3.1) and self-adaptive isotropic merging (Sec. 3.2), and demonstrate that jointly optimizing these two stages leads to more effective knowledge integration and retention in continual learning.

### 3.1 SHARPNESS-AWARE BLOCK COORDINATE DESCENT OPTIMIZER

To address parameter interference and catastrophic forgetting, we propose the **Sharpness-Aware Block Coordinate Descent (SA-BCD)** optimizer. SA-BCD enhances model adaptability and generalization through a dual selection mechanism: identifying task-sensitive parameters and exploring optimal update directions within the loss landscape. This strategy enables efficient adaptation while preserving general knowledge, mitigating parameter interference. By guiding the model towards flatter minima, SA-BCD creates an ideal foundation for subsequent merging.

Concretely, at each iteration, SA-BCD computes the gradient of the current parameters $\Theta_t$ with respect to the loss function $\mathcal{L}(\Theta_t, \mathcal{D})$, and updates the first-order momentum as follows:

$$m_t = \beta_1 m_{t-1} + (1 - \beta_1)\nabla_\Theta \mathcal{L}(\Theta_t, \mathcal{D}), \tag{7}$$

where $\beta_1$ is the first-order momentum coefficient ($\beta_1 = 0.9$ by default). The magnitude of $|m_t|$ is then used to select the top $p\%$ of parameters with the largest absolute values, denoted as $\Omega_t = \text{Top}_p(|m_t|)$ ($p = 30\%$ by default). These parameters are prioritized for updating, helping the model adapt to new tasks while minimizing interference with parameters related to general knowledge. For the selected subset $\Omega_t$, SA-BCD introduces a sharpness-aware perturbation mechanism to explore the worst-case direction in the loss landscape:

$$\epsilon_{t,\Omega}^* = \rho \frac{\nabla_\Theta \mathcal{L}(\Theta_t, \mathcal{D})_{\Omega_t}}{\|\nabla_\Theta \mathcal{L}(\Theta_t, \mathcal{D})_{\Omega_t}\|_2} \tag{8}$$

where $\rho$ is a hyperparameter controlling the perturbation magnitude, and $\nabla_\Theta \mathcal{L}(\Theta_t, \mathcal{D})_{\Omega_t}$ denotes the gradient restricted to the subset $\Omega_t$. This perturbation enables the optimizer to probe the geometry of the loss surface, steering the model away from sharp minima and towards flatter, more generalizable regions. Next, the optimizer updates the second-order momentum and applies bias correction as in Adam (Kingma & Ba, 2014): $v_t = \beta_2 v_{t-1} + (1 - \beta_2)(\nabla_\Theta \mathcal{L}(\Theta_t, \mathcal{D}))^2$, $\hat{m}_t = m_t/(1 - \beta_1^t)$, $\hat{v}_t = v_t/(1 - \beta_2^t)$. The gradient is then recomputed at the perturbed point $\Theta_t + \epsilon_{t,\Omega}^*$, yielding $g_t' = \nabla_\Theta \mathcal{L}(\Theta_t + \epsilon_{t,\Omega}^*, \mathcal{D})$. Finally, SA-BCD updates only the parameters in $\Omega_t$, while keeping the others unchanged. The update rule is:

$$\Theta_{t+1,i} = \begin{cases} \Theta_{t,i} - \eta \frac{\hat{m}_{t,i}}{\sqrt{\hat{v}_{t,i}} + \epsilon} g_{t,i}', & i \in \Omega_t \\ \Theta_{t,i}, & i \notin \Omega_t \end{cases} \tag{9}$$

where $\eta$ is the learning rate and $\epsilon$ is a small constant to prevent division by zero.

By computing gradients at the perturbed point (i.e., $\Theta_t + \epsilon_{t,\Omega}^*$) rather than the original point (i.e., $\Theta_t$), SA-BCD encourages convergence to flatter regions while preserving general knowledge. The complete algorithm is provided in Appendix E, with theoretical convergence guarantees in Appendix B.1.

### 3.2 Sharpness-Aware Isotropic Merging for Continual Learning

Building upon SA-BCD (Sec. 3.1), we present our complete approach to continual learning. SA-BCD guides models toward flatter minima and selectively updates task-sensitive parameters, creating parameter updates with reduced interference and enhanced flatness. To fully leverage these benefits, effective continual learning also requires optimized knowledge integration in the merging stage. Isotropic merging preserves this carefully structured knowledge and ensures balanced representation across all tasks, maximizing knowledge retention and subspace alignment.

We propose **Sharpness-Aware Isotropic Merging (SAIM)**, a continual learning model merging framework that integrates sharpness-aware fine-tuning and adaptive isotropic merging. SAIM maximizes knowledge retention and subspace alignment across tasks, addressing catastrophic forgetting and parameter interference.

For each new task, SAIM consists of two stages: (1) sharpness-aware fine-tuning using SA-BCD, and (2) adaptive isotropic merging in parameter space. Specifically, at each step $t$, given the current model parameters $\theta_{t-1}$ and a new task $T_t$, we first fine-tune the current merged model $\theta_{t-1}$ on $T_t$ using the SA-BCD optimizer (see Algorithm 1), obtaining the task expert model $\theta_{T_t}$ and defining the task vector $\Delta_{T_t} = \theta_{T_t} - \theta_{t-1}$.

During the merging stage, for each layer $k$, SAIM combines the cumulative update $\Delta_{cum}^k = \theta_{t-1}^k - \theta_{pre}^k$, which encodes all historical task knowledge, with the current task update $\Delta_{T_t}^k$ via a weighted sum to obtain the merged update:

$$\Delta_{com}^k = (1 + \lambda)\Delta_{cum}^k + (1 - \lambda)\Delta_{T_t}^k \tag{10}$$

where $\lambda$ is a coefficient balancing historical and new knowledge. In most cases, $\lambda$ is set to 0 for equal weighting.

Next, for each layer, SAIM performs singular value decomposition (SVD) on $\Delta_{com}^k$, i.e., $\Delta_{com}^k = U^k \Sigma^k (V^k)^\top$, where $U^k$ and $V^k$ are the left and right singular vectors, and $\Sigma^k$ is the diagonal matrix of singular values. To achieve isotropic merging and enhance subspace alignment, SAIM adaptively balances the singular value spectrum. Specifically, it first computes the mean singular value $\bar{\sigma}^k = \frac{1}{r} \sum_{i=1}^r \sigma_i^k$, and then constructs a new diagonal matrix by interpolating with the mean:

$$\hat{\Sigma}^k = \bar{\sigma}^k + (\Sigma^k - \bar{\sigma}^k) \times \frac{1}{\sqrt{t}} \tag{11}$$

where $t$ is the current task index. Notably, when the number of tasks is small, the singular value spectrum is typically dominated by a few large singular values, with the rest close to zero, indicating that parameter updates are concentrated in a few directions. In this case, the interpolation coefficient is large, which helps preserve the original spectrum and avoids degrading early task performance. As the number of tasks increases, the singular value spectrum gradually flattens, energy is distributed more evenly across directions, resulting in stronger isotropy and more balanced task representation in the merged model (Marczak et al., 2025). The merged update for each layer is reconstructed as $\Delta_{merged}^k = U^k \hat{\Sigma}^k (V^k)^\top$, and the final model parameters are updated as:

$$\theta_t = \theta_0 + \alpha \Delta_{merged} \tag{12}$$

where $\alpha$ is a scaling factor determined by validation set search. By repeating this process for each task, SAIM produces a final model $\theta_{final}$ that efficiently integrates multi-task knowledge and achieves high subspace alignment. The complete algorithmic procedure of SAIM is provided in Algorithm 2 in Appendix E.

## 4 Experiments

We evaluate our method on both vision and language continual learning tasks. Sec. 4.1 presents settings and results for vision tasks, while Sec. 4.2 covers large language model experiments. Sec. 4.3 analyzes the effectiveness of our approach. Due to page limitations, additional results and implementation details are available in the Appendix.

## 4.1 EXPERIMENTS ON VISION TASKS

Following the FusionBench protocol (Tang et al., 2024; Qiu et al., 2025), we evaluate our method on 20 diverse image classification tasks using three CLIP-ViT backbones (ViT-B/32, ViT-B/16, ViT-L/14). For each new task, the model is continually fine-tuned and merged, simulating a realistic continual learning scenario.

**Evaluation metrics.** We use average accuracy (ACC) and backward transfer (BWT, Lin et al., 2022) as the main evaluation metrics. ACC is defined as the mean accuracy over all tasks using the final merged model: $\text{ACC} = \frac{1}{T}\sum_{i=1}^{T} a_i(\theta_t^{merged})$, where $a_i(\cdot)$ denotes the accuracy on task $i$. BWT measures the extent of forgetting by comparing performance on earlier tasks before and after merging: $\text{BWT} = \frac{1}{T-1}\sum_{i=1}^{T-1}[a_i(\theta_t^{merged}) - a_i(\theta_i^{merged})]$.

Table 1: Comparative results of continual merging, reporting average accuracy (ACC) and backward transfer (BWT) over different numbers of tasks. Best results are in bold; second-best are underlined.

| Method | ViT-B/32 | | | ViT-B/16 | | | ViT-L/14 | | |
|---|---|---|---|---|---|---|---|---|---|
| | 8 tasks | 14 tasks | 20 tasks | 8 tasks | 14 tasks | 20 tasks | 8 tasks | 14 tasks | 20 tasks |
| Pre-Trained | 48.1 | 56.9 | 55.6 | 55.4 | 62.0 | 59.8 | 64.9 | 69.1 | 65.6 |
| Fine-Tuned | 90.4 | 89.3 | 89.8 | 92.4 | 91.3 | 91.6 | 94.3 | 93.4 | 93.5 |
| C. Fine-Tuned | 79.8 | 67.4 | 62.6 | 82.9 | 72.2 | 68.2 | 90.0 | 70.9 | 77.7 |
| **ACC (%) ↑** | | | | | | | | | |
| Average (SWA) | $66.3_{\pm0.0}$ | $65.4_{\pm0.0}$ | $61.1_{\pm0.0}$ | $72.3_{\pm0.0}$ | $69.7_{\pm0.0}$ | $64.8_{\pm0.0}$ | $80.0_{\pm0.0}$ | $77.5_{\pm0.0}$ | $71.1_{\pm0.0}$ |
| C. Task Arithmetic | $67.5_{\pm0.0}$ | $66.5_{\pm0.0}$ | $60.0_{\pm0.0}$ | $77.1_{\pm0.0}$ | $70.9_{\pm0.0}$ | $64.2_{\pm0.0}$ | $82.1_{\pm0.0}$ | $77.9_{\pm0.0}$ | $70.3_{\pm0.0}$ |
| C. TIES-Merging | $49.0_{\pm10.2}$ | $66.2_{\pm0.6}$ | $59.9_{\pm0.7}$ | $66.8_{\pm3.7}$ | $70.5_{\pm0.8}$ | $63.0_{\pm1.6}$ | $64.3_{\pm7.0}$ | $78.0_{\pm0.6}$ | $68.3_{\pm0.9}$ |
| MagMAX-IND | $70.7_{\pm0.0}$ | $67.0_{\pm0.0}$ | $61.2_{\pm0.0}$ | $76.7_{\pm1.8}$ | $67.0_{\pm0.0}$ | $62.5_{\pm0.0}$ | $83.4_{\pm0.0}$ | $71.2_{\pm0.0}$ | $71.2_{\pm0.0}$ |
| OPCM | $75.5_{\pm0.5}$ | $71.9_{\pm0.3}$ | $65.7_{\pm0.2}$ | $81.8_{\pm0.3}$ | $77.1_{\pm0.5}$ | $70.3_{\pm0.2}$ | $87.0_{\pm0.4}$ | $83.5_{\pm0.2}$ | $76.0_{\pm0.2}$ |
| EWC | $\mathbf{84.7}_{\pm1.3}$ | $74.1_{\pm1.2}$ | $67.1_{\pm0.7}$ | $\mathbf{87.8}_{\pm1.1}$ | $77.4_{\pm0.9}$ | $73.3_{\pm1.3}$ | $\mathbf{91.2}_{\pm0.6}$ | $85.0_{\pm0.9}$ | $\underline{82.9}_{\pm1.4}$ |
| ER | $81.8_{\pm1.9}$ | $\underline{74.6}_{\pm1.0}$ | $\underline{68.8}_{\pm1.5}$ | $\underline{87.7}_{\pm1.3}$ | $\underline{78.6}_{\pm0.7}$ | $\underline{73.4}_{\pm1.2}$ | $90.8_{\pm1.7}$ | $\underline{85.1}_{\pm0.7}$ | $80.9_{\pm1.3}$ |
| **SAIM (Ours)** | $\underline{82.1}_{\pm0.7}$ | $\mathbf{78.7}_{\pm0.5}$ | $\mathbf{73.6}_{\pm0.3}$ | $86.2_{\pm0.5}$ | $\mathbf{82.0}_{\pm0.5}$ | $\mathbf{79.0}_{\pm0.9}$ | $\underline{91.1}_{\pm0.1}$ | $\mathbf{88.5}_{\pm0.5}$ | $\mathbf{87.5}_{\pm0.5}$ |
| **BWT (%) ↑** | | | | | | | | | |
| Average (SWA) | $-11.5_{\pm2.2}$ | $-8.0_{\pm1.3}$ | $-7.1_{\pm2.1}$ | $-9.7_{\pm1.5}$ | $-7.1_{\pm1.4}$ | $-7.3_{\pm1.7}$ | $-7.3_{\pm1.4}$ | $-5.8_{\pm1.0}$ | $-6.4_{\pm1.5}$ |
| C. Task Arithmetic | $-9.6_{\pm1.5}$ | $\underline{-1.3}_{\pm1.6}$ | $\underline{-3.4}_{\pm1.0}$ | $-4.2_{\pm1.0}$ | $\underline{-1.3}_{\pm0.4}$ | $\underline{-3.6}_{\pm0.4}$ | $-7.1_{\pm0.8}$ | $\underline{-1.8}_{\pm0.3}$ | $\underline{-3.3}_{\pm0.3}$ |
| C. TIES-Merging | $-15.3_{\pm8.0}$ | $\mathbf{1.9}_{\pm0.6}$ | $\mathbf{-1.5}_{\pm0.7}$ | $-5.5_{\pm0.4}$ | $\mathbf{1.4}_{\pm0.7}$ | $\mathbf{-1.5}_{\pm1.2}$ | $-13.0_{\pm5.7}$ | $\mathbf{-1.1}_{\pm0.4}$ | $\mathbf{-2.9}_{\pm1.0}$ |
| MagMAX-IND | $-8.3_{\pm1.3}$ | $-7.4_{\pm1.4}$ | $-7.2_{\pm1.6}$ | $-6.1_{\pm1.3}$ | $-7.4_{\pm2.0}$ | $-8.0_{\pm2.2}$ | $-5.0_{\pm0.8}$ | $-6.0_{\pm2.1}$ | $-6.5_{\pm2.1}$ |
| OPCM | $-6.3_{\pm1.1}$ | $-6.0_{\pm1.0}$ | $-7.8_{\pm1.5}$ | $-4.8_{\pm0.7}$ | $-5.1_{\pm1.4}$ | $-6.3_{\pm2.2}$ | $\underline{-2.6}_{\pm1.0}$ | $-4.3_{\pm0.7}$ | $-6.5_{\pm1.8}$ |
| EWC | $\underline{-4.7}_{\pm0.9}$ | $-16.1_{\pm1.5}$ | $-23.7_{\pm1.2}$ | $\underline{-3.7}_{\pm0.4}$ | $-14.5_{\pm1.3}$ | $-20.0_{\pm1.1}$ | $-2.9_{\pm0.2}$ | $-9.1_{\pm0.7}$ | $-11.8_{\pm1.3}$ |
| ER | $-7.8_{\pm2.3}$ | $-15.0_{\pm1.4}$ | $-21.6_{\pm1.7}$ | $-3.9_{\pm0.3}$ | $-13.2_{\pm0.8}$ | $-19.1_{\pm1.2}$ | $-2.9_{\pm0.1}$ | $-8.6_{\pm0.6}$ | $-13.4_{\pm0.9}$ |
| **SAIM (Ours)** | $\mathbf{-3.9}_{\pm0.8}$ | $-5.7_{\pm0.6}$ | $-11.5_{\pm0.5}$ | $\mathbf{-2.9}_{\pm0.4}$ | $-4.8_{\pm1.0}$ | $-9.0_{\pm1.1}$ | $\mathbf{-1.1}_{\pm0.2}$ | $-2.5_{\pm0.7}$ | $-4.3_{\pm0.7}$ |

**Main results.** Table 1 compares our method with various continual learning and model merging methods. In terms of ACC, SAIM consistently outperforms other methods as the number of tasks increases, especially at 14 and 20 tasks. For example, on ViT-L/14 with 20 tasks, SAIM achieves 87.5%, which is 10 percentage points higher than OPCM (76.0%). While EWC and ER have a slight advantage at 8 tasks, SAIM shows stronger scalability and stability as the number of tasks increases. For backward transfer (BWT), SAIM achieves results comparable to mainstream methods. This is because SAIM effectively suppresses interference from previous tasks when merging new tasks, resulting in a relatively high initial accuracy for new tasks. Consequently, although there may be some subsequent decline, the overall change is moderate.

## 4.2 EXPERIMENTS ON LARGE LANGUAGE MODELS

We conducted continual learning experiments on large language models using both the MergeBench (He et al., 2025) and TRACEBench (Wang et al., 2023) benchmarks to evaluate the effectiveness of the SAIM method in merging domain-specialized LLMs and continual adaptation to diverse language tasks.

**Evaluation metrics.** In the MergeBench experiments, we report the average score across five domains (instruction, math, multilingual, coding, safety). Each domain score is obtained by directly averaging the core metrics (such as accuracy, pass@1, etc.) of the main tasks in that domain (He et al., 2025), and the final overall score is the arithmetic mean of the five domain scores. In the TRACEBench experiments, we report the arithmetic mean of the main metric scores across all tasks. The specific metrics used for calculation are detailed in the Appendix C.1.

Table 2: Comparison of Model Merging Methods on MergeBench with Llama-3.2-3B.

| Method | Task Performance (Average Score) | | | | | Average Score |
|--------|-------------|------|--------------|--------|--------|---------------|
| | Instruction | Math | Multilingual | Coding | Safety | |
| Pre-Trained | 0.0998 | 0.3078 | 0.4531 | 0.2675 | 0.2856 | 0.2828 |
| Fine-Tuned | 0.3503 | 0.5777 | 0.4608 | 0.4420 | 0.7535 | 0.5129 |
| SWA | 0.1322 | 0.4245 | **0.4580** | 0.3754 | 0.3771 | 0.3535 |
| TIES-Merging | 0.1682 | 0.4488 | 0.4523 | 0.4004 | 0.4118 | 0.3763 |
| DARE | 0.2643 | 0.4405 | 0.4399 | 0.4050 | 0.4396 | 0.3979 |
| MagMAX-IND | 0.2791 | 0.4928 | 0.4246 | 0.3913 | 0.4019 | 0.3980 |
| Task Arithmetic | 0.2699 | 0.4610 | 0.4423 | **0.4183** | 0.4445 | 0.4072 |
| **SAIM (Ours)** | **0.3872** | **0.5830** | 0.4332 | 0.3803 | **0.4848** | **0.4537** |

Table 3: Comparison of Model Merging Methods on MergeBench with Llama-3.1-8B (Excluding Math).

| Method | Task Performance (Average Score) | | | | Average Score |
|--------|-------------|--------------|--------|--------|---------------|
| | Instruction | Multilingual | Coding | Safety | |
| Pre-Trained | 0.0795 | 0.5324 | 0.3631 | 0.3698 | 0.3362 |
| Fine-Tuned | 0.3752 | 0.5395 | 0.5627 | 0.7741 | 0.5629 |
| TIES-Merging | 0.0619 | 0.4865 | 0.3712 | 0.7891 | 0.4272 |
| DARE | 0.2006 | 0.3206 | 0.4393 | 0.7601 | 0.4302 |
| SWA | 0.0277 | **0.5309** | 0.4679 | 0.7582 | 0.4462 |
| MagMAX-IND | 0.1248 | 0.4077 | 0.5107 | **0.8414** | 0.4711 |
| Task Arithmetic | **0.2052** | 0.3770 | 0.5190 | 0.8018 | 0.4757 |
| **SAIM (Ours)** | 0.2043 | 0.4712 | **0.5202** | 0.7790 | **0.4937** |

**Main results.** As shown in Tables 2 and 3, SAIM achieved the best average scores on both Llama-3.2-3B and Llama-3.1-8B (excluding Math due to checkpoint issues[1]). For example, on Llama-3.2-3B, SAIM achieved an average score of 0.4537, significantly higher than Task Arithmetic (0.4072). On Llama-3.1-8B (excluding Math), SAIM reached 0.4937, outperforming Task Arithmetic (0.4757) and other baselines. This demonstrates SAIM's strong generalization and knowledge integration capability in large-scale LLM merging scenarios. On TRACEBench (Table 4), SAIM also achieved the highest average score (0.5468) with the 3B model, significantly outperforming other baseline methods such as DARE (0.4906). Similar trends were observed with the 1B model (see Appendix D.7).

Table 4: Comparison of Model Merging Methods on TraceBench with Llama-3.2-3B-Instruct.

| Method | Task Performance | | | | | | | Average Score |
|--------|----------|------|-------------|-----------|------------|------------|----------|---------------|
| | C-STANCE | FOMC | MeetingBank | ScienceQA | NumGLUE-cm | NumGLUE-ds | 20Minuten | |
| Pre-Trained | 0.4082 | 0.3528 | 0.2054 | 0.8962 | 0.1707 | 0.2195 | 0.3857 | 0.3770 |
| Fine-Tuned | 0.5415 | 0.6835 | 0.4317 | 0.9335 | 0.6098 | 0.6463 | 0.3898 | 0.6057 |
| SWA | 0.4617 | 0.5665 | 0.2213 | 0.9140 | 0.4390 | 0.4146 | 0.3891 | 0.4866 |
| Task Arithmetic | 0.4685 | 0.5605 | 0.2186 | 0.9100 | 0.4634 | 0.4024 | 0.3862 | 0.4871 |
| MagMAX-IND | **0.4820** | 0.5202 | 0.2340 | 0.9055 | 0.4634 | 0.4207 | 0.3912 | 0.4881 |
| TIES-Merging | 0.4670 | 0.5706 | 0.2217 | 0.9130 | 0.4634 | 0.4024 | 0.3906 | 0.4898 |
| DARE | 0.4630 | **0.5867** | 0.2203 | 0.9055 | 0.4634 | 0.4085 | 0.3871 | 0.4906 |
| **SAIM (Ours)** | 0.4750 | 0.5685 | **0.2491** | **0.9240** | **0.6098** | **0.6098** | **0.3914** | **0.5468** |

## 4.3 EMPIRICAL AND THEORETICAL ANALYSIS

To understand our method's effectiveness, we analyze it from two key perspectives: parameter change magnitude and weight disentanglement.

**Parameter Change Magnitude Analysis.** We measure parameter changes after fine-tuning on five visual task categories and compare SA-BCD with standard Adam optimizer. As shown in Fig. 2,

---

[1]Math task is excluded because the official MergeBench Llama-3.1-8B checkpoint for Math shows lower performance than the pre-trained model (0.4026 vs. 0.5625) on evaluation. Results including all five tasks (with Math) are available in Appendix D.6.

parameter changes after SA-BCD fine-tuning concentrate around $10^{-4.5}$, significantly smaller than those under Adam (around $10^{-3.5}$). This indicates SA-BCD causes less disturbance to pre-trained parameters, better preserving generalization ability (Chen et al., 2024) and reducing interference with original knowledge.

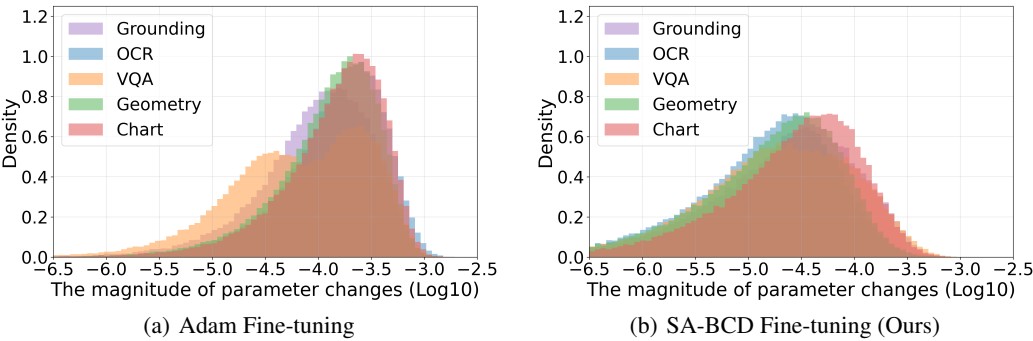

(a) Adam Fine-tuning         (b) SA-BCD Fine-tuning (Ours)

Figure 2: Comparison of parameter change distributions under different fine-tuning methods. This figure shows the distribution of parameter changes for five task categories after fine-tuning with (a) Adam and (b) our SA-BCD optimizer. It can be seen that SA-BCD results in overall smaller parameter changes, which helps preserve the generalization ability of the pre-trained model.

**Weight Disentanglement Visualization.** We quantitatively analyze the output differences between the merged model and each task-specific model on their respective tasks. Let $\boldsymbol{\theta}_0$ denote pre-trained model parameters, $\boldsymbol{\tau}_t$ the task vector for task $t$, and $\alpha_t$ the task coefficient. The weight disentanglement error is defined as:

$$\xi(\alpha_1, \alpha_2) = \sum_{t=1}^{2} \mathbb{E}_{\boldsymbol{x} \in X^{(t)}} \left[ \text{dist}\big( f(\boldsymbol{x}; \boldsymbol{\theta}_0 + \alpha_t \boldsymbol{\tau}_t), \ f(\boldsymbol{x}; \boldsymbol{\theta}_{merge}) \big) \right] \tag{13}$$

where $f(\cdot)$ denotes model output, $\text{dist}(\cdot, \cdot)$ is output distance metric, and $X^{(t)}$ is data for task $t$. Lower $\xi$ indicates less interference between tasks.

By visualizing the heatmaps of $\xi$ under different $\alpha_1$ and $\alpha_2$ (see Fig. 3), we demonstrate the degree of weight disentanglement of the merged model. Experimental results show that the merged model under SA-BCD fine-tuning achieves lower disentanglement errors in the two-task scenario, indicating that our method effectively reduces parameter interference. Further analysis of multi-task scenarios and subspace alignment with increasing task numbers is provided in Appendix D.3.

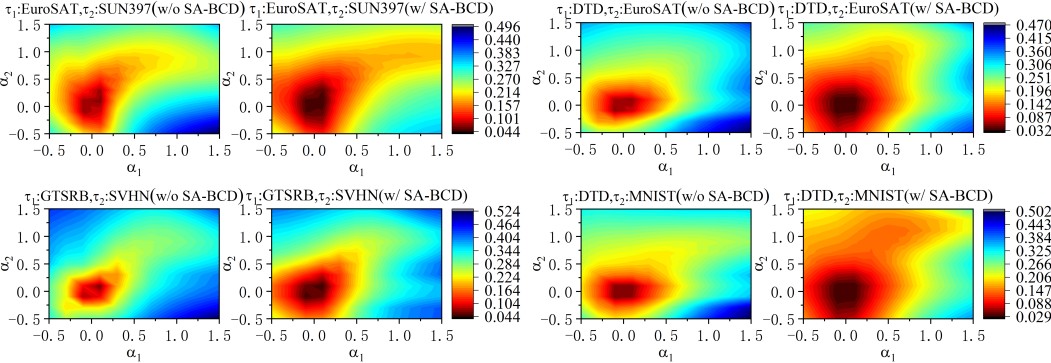

Figure 3: Visualization of weight disentanglement error in the two-task merging scenario. Each subplot shows the output differences between the merged model and each task-specific model on their respective tasks under different fine-tuning methods (e.g., Adam and SA-BCD). Darker colors indicate lower disentanglement error, reflecting better weight disentanglement.

## 5 RELATED WORK

**Continual Learning (CL).** Continual learning aims to enable models to accumulate knowledge from a stream of new tasks. Traditional approaches such as regularization constraints (EWC, Kirkpatrick et al., 2017), structural expansion (modular networks, Gurbuz & Dovrolis, 2022), and experience replay (Buzzega et al., 2020) can alleviate forgetting to some extent, but often require storing historical data or introducing additional parameters, making it difficult to balance stability and plasticity. Recent methods such as MoFO (Chen et al., 2024), which selectively updates parameters via momentum filtering, and PAM (Sokar et al., 2025), which improves merging through parameter alignment, have made progress in optimizing efficiency and knowledge retention. However, most of these methods treat fine-tuning and knowledge integration as separate processes, failing to fully exploit their synergy. Our method is based on the idea of joint fine-tuning and merging optimization, aiming to achieve efficient knowledge acquisition and retention in continual learning scenarios.

**Model Merging.** Model merging has become a key research direction in multi-task and continual learning (Yang et al., 2024). Early approaches such as parameter averaging (Izmailov et al., 2018; Wortsman et al., 2022) and task arithmetic (Ilharco et al., 2022) integrate knowledge through simple parameter operations, but often suffer performance degradation in multi-task scenarios. To address parameter conflicts, TIES (Yadav et al., 2023) reduces sign conflicts to improve merging, while MagMAX (Marczak et al., 2024) adopts a maximum magnitude selection strategy to enhance stability. Recently, ISO-C (Marczak et al., 2025) has significantly improved inter-task alignment and merging performance by flattening the singular value spectrum and introducing task-specific subspaces. However, existing methods still face knowledge conflicts between tasks in continual learning, limiting the performance of merged models. Our method employs adaptive isotropic merging to dynamically balance the singular value spectrum, thereby enhancing the knowledge integration and generalization ability of merged models.

**Sparsification & Sharpness-Aware Optimization.** Recent studies have demonstrated that parameters are highly redundant during fine-tuning, and sparsification or parameter selection strategies (e.g., DARE Yu et al., 2024) can effectively alleviate parameter interference in model merging. DARE discards most delta parameters and rescales the remaining ones, enabling efficient multi-task integration. In a parallel line of research, sharpness-aware minimization (SAM Foret et al., 2021) guides models to converge to flatter minima, improving generalization and stability. Recently, SAFT Lee et al., 2025 applied sharpness-aware optimization to model merging scenarios to reduce parameter interference between task-specific models. Unlike SAFT which only addresses sharpness in isolated multi-task settings, our approach operates in continual learning settings without historical data access and evaluates on both vision and language tasks. Inspired by these works, we introduce a sharpness-aware block coordinate descent (SA-BCD) optimizer that jointly leverages parameter selection and flat minima optimization to facilitate efficient merging and continual learning.

## 6 CONCLUSION

We present Sharpness-Aware Isotropic Merging (SAIM), a framework that effectively combines sharpness-aware fine-tuning and adaptive isotropic merging techniques. Our experimental results demonstrate that SAIM exhibits excellent performance across both vision and language tasks, not only achieving significant improvements on individual tasks but also maintaining stable performance as the number of tasks increases. Through the synergistic effect of the Sharpness-Aware Block Coordinate Descent optimizer and adaptive isotropic merging algorithm, SAIM has made notable progress in cross-task knowledge integration and retention, providing a new solution approach for the continual learning domain.

**Limitations.** Despite SAIM's strong performance in continual learning, there are still some limitations worth noting. First, our current experimental settings primarily focus on classification and language modeling tasks, and future work could explore more diverse task types; second, this study only explores single-modality input scenarios (pure vision or pure language), without addressing knowledge integration and transfer in multi-modal learning environments, which has significant importance in real-world applications; additionally, the model's adaptability under certain extreme data distributions requires further validation. Improvements in these areas will be directions for our future work, to enhance SAIM's applicability and effectiveness in complex scenarios.

## ETHICS STATEMENT

This research focuses on developing model merging techniques for efficient knowledge integration in AI systems. We acknowledge that all machine learning methods, including ours, may inherit biases from training data. While our primary contributions are algorithmic and methodological in nature, we encourage practitioners to evaluate potential societal impacts before deployment in sensitive applications. Our work aims to advance technology, and we urge users to use our technology responsibly, complying with relevant laws and regulations.

## REPRODUCIBILITY STATEMENT

All experimental details necessary to reproduce our results are documented in Appendix C. Our implementation, including the SAIM framework and evaluation protocols, is publicly available at `https://anonymous.4open.science/r/SAIM-Continual-Learning-406E`.

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

APPENDIX

This appendix provides supplementary material to support our main findings. It consists of five main sections: (A) LLM Usage Statement, (B) Proofs presenting theoretical foundations of our approach, (C) Experimental Details describing implementation specifics, (D) Additional Results offering extended empirical evaluations, and (E) Algorithmic Details with full pseudocode for our core methods.

## A  LLM USAGE STATEMENT

This paper made use of a large language model (ChatGPT) exclusively for language polishing, spelling correction, and grammar checking. The LLM was not involved in literature retrieval or in the development of specific ideas. Following the polishing process, the authors carefully reviewed and revised the content as necessary and assume full responsibility for the final published version.

## B  PROOFS

In this section, we provide theoretical analyses for the key components of our SAIM approach. We present the convergence analysis of SA-BCD (B.1), examine the subspace alignment properties of isotropic merging (B.2), and analyze how our method reduces parameter interference in continual learning scenarios (B.3).

### B.1  CONVERGENCE ANALYSIS OF SA-BCD

We first analyze the convergence properties of our Sharpness-Aware Block Coordinate Descent (SA-BCD) optimizer, which selectively updates parameters based on momentum magnitude while incorporating sharpness-aware perturbations.

**Theorem 1** (Convergence of SA-BCD). Consider a loss function $\mathcal{L}(\theta)$ that is $\mu$-strongly convex and has $L$-Lipschitz continuous gradients. Let $\theta^*$ denote the global minimizer of $\mathcal{L}$. For the SA-BCD algorithm with parameter selection ratio $p$ and perturbation magnitude $\rho < \frac{1}{2L}$, after $T$ iterations, the expected optimality gap satisfies:

$$\mathbb{E}[\mathcal{L}(\theta_T) - \mathcal{L}(\theta^*)] \leq \left(1 - \frac{p\mu}{L}\right)^T [\mathcal{L}(\theta_0) - \mathcal{L}(\theta^*)] + \frac{2L\rho^2\sigma^2}{p\mu} \tag{14}$$

where $\sigma^2$ bounds the variance of the stochastic gradients.

**Proof.** Let $\Omega_t$ be the set of indices corresponding to the top $p\%$ of parameters with the largest momentum magnitude at iteration $t$. Let $g_t = \nabla_\theta \mathcal{L}(\theta_{t-1})$ be the full gradient and $\epsilon_t^*$ be the sharpness-aware perturbation.

Due to the $L$-Lipschitz continuity of the gradients, we have:

$$\mathcal{L}(\theta_t) \leq \mathcal{L}(\theta_{t-1}) + \langle \nabla_\theta \mathcal{L}(\theta_{t-1}), \theta_t - \theta_{t-1} \rangle + \frac{L}{2} \|\theta_t - \theta_{t-1}\|^2 \tag{15}$$

Since SA-BCD only updates parameters in $\Omega_t$, we can write:

$$\theta_t - \theta_{t-1} = -\eta \cdot \mathbf{1}_{\Omega_t} \odot \nabla_\theta \mathcal{L}(\theta_{t-1} + \epsilon_t^*) \tag{16}$$

where $\mathbf{1}_{\Omega_t}$ is the indicator vector for the selected parameters, and $\odot$ denotes element-wise multiplication.

Using the definition of the sharpness-aware perturbation $\epsilon_t^*$, and the properties of the momentum-based selection, we can show that:

$$\mathbb{E}[\langle \nabla_\theta \mathcal{L}(\theta_{t-1}), \mathbf{1}_{\Omega_t} \odot \nabla_\theta \mathcal{L}(\theta_{t-1} + \epsilon_t^*) \rangle] \geq p \cdot \|\nabla_\theta \mathcal{L}(\theta_{t-1})\|^2 - 2L\rho \|\nabla_\theta \mathcal{L}(\theta_{t-1})\|^2 \tag{17}$$

By selecting an appropriate learning rate $\eta = \frac{1}{L}$ and using the fact that $\rho < \frac{1}{2L}$, we obtain:

$$\mathbb{E}[\mathcal{L}(\theta_t)] \leq \mathcal{L}(\theta_{t-1}) - \frac{p}{2L} \|\nabla_\theta \mathcal{L}(\theta_{t-1})\|^2 + L\rho^2 \sigma^2 \tag{18}$$

By the $\mu$-strong convexity of $\mathcal{L}$, we have:

$$\|\nabla_\theta \mathcal{L}(\theta_{t-1})\|^2 \geq 2\mu[\mathcal{L}(\theta_{t-1}) - \mathcal{L}(\theta^*)] \tag{19}$$

Combining these inequalities and applying them recursively, we arrive at:

$$\mathbb{E}[\mathcal{L}(\theta_T) - \mathcal{L}(\theta^*)] \leq \left(1 - \frac{p\mu}{L}\right)^T [\mathcal{L}(\theta_0) - \mathcal{L}(\theta^*)] + \frac{2L\rho^2 \sigma^2}{p\mu} \qquad \square$$

### B.2 SUBSPACE ALIGNMENT ANALYSIS OF ISOTROPIC MERGING

Next, we analyze how isotropic merging enhances subspace alignment between task-specific and merged model representations.

**Theorem 2** (Subspace Alignment Enhancement via Isotropic Merging). Let $\Delta_t \in \mathbb{R}^{d \times d}$ be the parameter update matrix for task $t$, and $\Delta_{\text{TA}} = \sum_{t=1}^T \Delta_t$ be the task arithmetic merged matrix. Denote $\Delta_{\text{Iso}} = U\bar{\sigma}V^\top$ as the isotropically merged matrix, where $U$ and $V$ are the left and right singular vectors of $\Delta_{\text{TA}}$, and $\bar{\sigma}$ is the mean singular value. For any task $t$, the subspace alignment ratio satisfies:

$$\text{SAR}(\Delta_t, \Delta_{\text{Iso}}; k) \geq \text{SAR}(\Delta_t, \Delta_{\text{TA}}; k) + \gamma_t \tag{20}$$

where $\gamma_t \geq 0$ represents the alignment improvement, with equality if and only if the singular values of $\Delta_{\text{TA}}$ are already uniform.

**Proof.** Let $\Delta_{\text{TA}} = U\Sigma V^\top$ be the singular value decomposition of the task arithmetic merged matrix, where $\Sigma = \text{diag}(\sigma_1, \sigma_2, \ldots, \sigma_r)$ with $\sigma_1 \geq \sigma_2 \geq \ldots \geq \sigma_r$. The isotropically merged matrix is $\Delta_{\text{Iso}} = U\bar{\sigma}IV^\top$, where $\bar{\sigma} = \frac{1}{r}\sum_{i=1}^r \sigma_i$ and $I$ is the identity matrix.

The subspace alignment ratio for task $t$ with respect to the merged matrix $\Delta_M$ (either $\Delta_{\text{TA}}$ or $\Delta_{\text{Iso}}$) is defined as:

$$\text{SAR}(\Delta_t, \Delta_M; k) = \frac{\|\pi_{k,M}\Delta_t\|_F}{\|\Delta_t\|_F} \tag{21}$$

where $\pi_{k,M} = U_{k,M}U_{k,M}^\top$ is the projection operator onto the subspace spanned by the top $k$ left singular vectors of $\Delta_M$.

For both $\Delta_{\text{TA}}$ and $\Delta_{\text{Iso}}$, the top $k$ left singular vectors are identical (both are $U_{k,\text{TA}} = U_{k,\text{Iso}} = U_k$). However, when we calculate the projection, we need to account for the weighting of these vectors by the singular values.

For task arithmetic, the projection is:

$$\pi_{k,\text{TA}}\Delta_t = \sum_{i=1}^{k} \frac{\sigma_i}{\sum_{j=1}^{k} \sigma_j} \cdot (u_i^\top \Delta_t) \cdot u_i \tag{22}$$

For isotropic merging, the projection is:

$$\pi_{k,\text{Iso}}\Delta_t = \sum_{i=1}^{k} \frac{\bar{\sigma}}{\sum_{j=1}^{k} \bar{\sigma}} \cdot (u_i^\top \Delta_t) \cdot u_i = \sum_{i=1}^{k} \frac{1}{k} \cdot (u_i^\top \Delta_t) \cdot u_i \tag{23}$$

Let $\alpha_i = u_i^\top \Delta_t$. The squared Frobenius norm of the projections can be written as:

$$\|\pi_{k,\text{TA}}\Delta_t\|_F^2 = \sum_{i=1}^{k} \left( \frac{\sigma_i}{\sum_{j=1}^{k} \sigma_j} \right)^2 \cdot \|\alpha_i\|^2 \tag{24}$$

$$\|\pi_{k,\text{Iso}}\Delta_t\|_F^2 = \sum_{i=1}^{k} \left( \frac{1}{k} \right)^2 \cdot \|\alpha_i\|^2 = \frac{1}{k^2} \sum_{i=1}^{k} \|\alpha_i\|^2 \tag{25}$$

By the Cauchy-Schwarz inequality, and considering that the singular values are typically skewed (with a few large values and many small ones), we can show that:

$$\|\pi_{k,\text{Iso}}\Delta_t\|_F^2 - \|\pi_{k,\text{TA}}\Delta_t\|_F^2 = \sum_{i=1}^{k} \left[ \left( \frac{1}{k} \right)^2 - \left( \frac{\sigma_i}{\sum_{j=1}^{k} \sigma_j} \right)^2 \right] \cdot \|\alpha_i\|^2 \geq 0 \tag{26}$$

Therefore, we have:

$$\text{SAR}(\Delta_t, \Delta_{\text{Iso}}; k) = \frac{\|\pi_{k,\text{Iso}}\Delta_t\|_F}{\|\Delta_t\|_F} \geq \frac{\|\pi_{k,\text{TA}}\Delta_t\|_F}{\|\Delta_t\|_F} = \text{SAR}(\Delta_t, \Delta_{\text{TA}}; k) \qquad \square$$

Let $\gamma_t = \text{SAR}(\Delta_t, \Delta_{\text{Iso}}; k) - \text{SAR}(\Delta_t, \Delta_{\text{TA}}; k) \geq 0$, which represents the improvement in alignment due to isotropic merging.

## B.3 ANALYSIS OF PARAMETER INTERFERENCE REDUCTION

Finally, we analyze how SAIM reduces parameter interference between tasks in continual learning scenarios.

**Theorem 3** (SAIM Reduces Parameter Interference). Models fine-tuned using SA-BCD and merged via isotropic merging exhibit enhanced joint-task loss linearity and reduced parameter interference compared to standard optimization and merging methods. Specifically, for tasks $s$ and $t$ with datasets $\mathcal{D}_s$ and $\mathcal{D}_t$, and models with parameters $\theta_s^{SAIM}$ and $\theta_t^{SAIM}$ fine-tuned using SAIM, and $\theta_s^{SGD}$ and $\theta_t^{SGD}$ fine-tuned using standard SGD, we have:

$$|\delta_{SAIM}| \leq |\delta_{SGD}| \tag{27}$$

where $\delta_{method}$ represents the deviation from perfect linearity in the joint-task loss.

**Proof.** Consider the joint-task loss for a merged model $\theta_{merge} = \alpha\theta_s + (1-\alpha)\theta_t$ evaluated on the combined dataset $\mathcal{D} = \mathcal{D}_s \cup \mathcal{D}_t$:

$$\mathcal{L}_{JTL}(\theta_{merge}; \mathcal{D}) = \mathbb{E}_{(x,y)\sim\mathcal{D}}\mathcal{L}(f(x; \theta_{merge}), y) \tag{28}$$

The deviation from perfect linearity is defined as:

$$\delta = \mathcal{L}_{JTL}(\theta_{merge}; \mathcal{D}) - [\alpha\mathcal{L}_{JTL}(\theta_s; \mathcal{D}) + (1-\alpha)\mathcal{L}_{JTL}(\theta_t; \mathcal{D})] \tag{29}$$

Applying a second-order Taylor expansion of $\mathcal{L}_{JTL}$ around $\theta_s$ and $\theta_t$, we have:

$$\mathcal{L}_{JTL}(\theta_{merge}; \mathcal{D}) \approx \alpha\mathcal{L}_{JTL}(\theta_s; \mathcal{D}) + (1-\alpha)\mathcal{L}_{JTL}(\theta_t; \mathcal{D}) + \frac{1}{2}\alpha(1-\alpha)(\theta_t-\theta_s)^\top [H_s + H_t] (\theta_t-\theta_s) \tag{30}$$

where $H_s = \nabla^2 \mathcal{L}_{JTL}(\theta_s; \mathcal{D}_s)$ and $H_t = \nabla^2 \mathcal{L}_{JTL}(\theta_t; \mathcal{D}_t)$ are the Hessians at $\theta_s$ and $\theta_t$ respectively.

Thus, the deviation $\delta$ can be approximated as:

$$\delta \approx \frac{1}{2}\alpha(1-\alpha)(\theta_t - \theta_s)^\top (H_s + H_t)(\theta_t - \theta_s) \tag{31}$$

Taking the absolute value and using the spectral norm, we obtain the upper bound:

$$|\delta| \leq \frac{1}{2}\alpha(1-\alpha)\left[\lambda_{\max}(H_s) + \lambda_{\max}(H_t)\right] \|\theta_t - \theta_s\|^2 \tag{32}$$

Now, we analyze the effect of SAIM:

- **Flatter minima:** By Theorem 1, SA-BCD leads to flatter minima, i.e., $\lambda_{\max}(H_s^{SAIM}) \leq \lambda_{\max}(H_s^{SGD})$ and $\lambda_{\max}(H_t^{SAIM}) \leq \lambda_{\max}(H_t^{SGD})$.

- **Improved subspace alignment:** By Theorem 2, isotropic merging reduces the effective distance between task-specific solutions in the relevant subspace, i.e., $\|\theta_t^{SAIM} - \theta_s^{SAIM}\|^2 \leq \|\theta_t^{SGD} - \theta_s^{SGD}\|^2$.

Combining these two effects, for any $\alpha \in [0, 1]$, we have:

$$|\delta_{SAIM}| \leq \frac{1}{2}\alpha(1-\alpha)\left[\lambda_{\max}(H_s^{SAIM}) + \lambda_{\max}(H_t^{SAIM})\right] \|\theta_t^{SAIM} - \theta_s^{SAIM}\|^2 \tag{33}$$

$$|\delta_{SGD}| \leq \frac{1}{2}\alpha(1-\alpha)\left[\lambda_{\max}(H_s^{SGD}) + \lambda_{\max}(H_t^{SGD})\right] \|\theta_t^{SGD} - \theta_s^{SGD}\|^2 \tag{34}$$

Since both the Hessian terms and the parameter distance are smaller for SAIM, it follows that:

$$|\delta_{SAIM}| \leq |\delta_{SGD}| \qquad \qquad \square$$

This means that models trained and merged using SAIM exhibit less deviation from linearity in their joint-task loss, indicating reduced parameter interference between tasks.

The three theorems collectively demonstrate that SAIM effectively addresses the core challenges in continual learning by: (1) ensuring proper convergence to flatter minima via SA-BCD, (2) enhancing subspace alignment via isotropic merging, and (3) reducing parameter interference between tasks, leading to improved performance retention across multiple tasks.

## C  EXPERIMENTAL DETAILS

This section outlines the experimental setup used to evaluate our method. We describe datasets and task configurations (C.1), implementation details (C.2), baseline methods (C.3), parameter change analysis (C.4), and weight disentanglement visualization (C.5), ensuring reproducibility and providing context for our empirical findings.

### C.1  DATASETS AND TASK SETTINGS

**Vision Tasks.** For evaluating SAIM in the vision domain, we follow the FusionBench protocol (Tang et al., 2024; Qiu et al., 2025), using 20 diverse image classification datasets organized into three task groups. To ensure robustness and consistency, we conduct experiments with five different random task permutations (using seeds 42-46), as shown in Table 6. For clarity, we denote the datasets by their indices: (1) SUN397(Xiao et al., 2010), (2) Stanford Cars(Krause et al., 2013), (3) RESISC45(Cheng et al., 2017), (4) EuroSAT(Helber et al., 2019), (5) SVHN(Netzer et al., 2011), (6) GTSRB(Stallkamp et al., 2012), (7) MNIST(LeCun et al., 2002), (8) DTD(Cimpoi et al., 2014), (9) Flowers102(Nilsback & Zisserman, 2008), (10) PCAM(Veeling et al., 2018), (11) FER2013(Goodfellow et al., 2013), (12) Oxford-IIIT Pets(Parkhi et al., 2012), (13) STL-10(Coates et al., 2011), (14) CIFAR-100 and (15) CIFAR-10(Krizhevsky et al., 2009), (16) Food-101(Bossard et al., 2014), (17) Fashion-MNIST(Xiao et al., 2017), (18) EMNIST(Cohen et al., 2017), (19) KM-NIST(Clanuwat et al., 2018), (20) Rendered SST-2(Socher et al., 2013).

- **8-task setting**: SUN397, Stanford Cars, RESISC45, EuroSAT, SVHN, GTSRB, MNIST, and DTD

- **14-task setting**: The 8-task setting plus Flowers102, PCAM, FER2013, Oxford-IIIT Pets, STL-10, and CIFAR-100

- **20-task setting**: The 14-task setting plus CIFAR-10, Food-101, Fashion-MNIST, EMNIST, KMNIST, and Rendered SST-2

Table 5: Extended downstream datasets used in our experiments.

| Dataset | #Classes | #Train (k) | #Test (k) | Task |
|---|---|---|---|---|
| SUN397 | 287 | 19.9 | 19.9 | Scene category |
| Stanford Cars | 196 | 8.1 | 8.0 | Car series |
| RESISC45 | 45 | 18.9 | 6.3 | Remote-sensing scene |
| EuroSAT | 17 | 21.6 | 2.7 | Satellite land-use |
| SVHN | 10 | 73.3 | 26.0 | Digit recognition |
| GTSRB | 43 | 39.2 | 12.6 | Traffic sign |
| MNIST | 10 | 60 | 10 | Hand-written digit |
| DTD | 47 | 3.8 | 1.9 | Texture recognition |
| Flowers102 | 102 | 1.0 | 6.1 | Flower species |
| PCAM | 2 | 262 | 32.8 | Tumour classification |
| FER2013 | 7 | 28.7 | 3.6 | Facial emotion |
| Oxford-IIIT Pets | 37 | 3.7 | 3.7 | Animal species |
| STL-10 | 10 | 5 | 8 | Object recognition |
| CIFAR-100 | 100 | 50 | 10 | Natural object |
| CIFAR-10 | 10 | 50 | 10 | Natural object |
| Food-101 | 101 | 75.8 | 25.3 | Food type |
| Fashion-MNIST | 10 | 60 | 10 | Fashion product |
| EMNIST | 10 | 60 | 10 | Hand-written digit |
| KMNIST | 10 | 60 | 10 | Kuzushiji character |
| Rendered SST-2 | 2 | 6.9 | 1.8 | Rendered sentiment |

These datasets span various domains including scene recognition, object classification, remote sensing, texture analysis, and character recognition. To provide a comprehensive overview, we list the detailed statistics of all downstream datasets used in our experiments in Table 5, following prior works Tang et al., 2024; Qiu et al., 2025. The table includes the number of classes, training and test samples, and the corresponding task type for each dataset.

To ensure robustness to task ordering effects, we conduct experiments with five different random task permutations, as shown in Table 6. For all vision experiments, we use CLIP-ViT models (ViT-B/32, ViT-B/16, and ViT-L/14) as the backbone architectures.

Table 6: The five different task orderings used in our vision experiments.

| Order | Dataset Order by ID |
|---|---|
| 1 | (6→5→7→2→3→8→1→4→10→14→13→11→12→9→20→15→16→19→18→17) |
| 2 | (7→8→5→4→2→6→3→1→13→12→9→14→10→11→15→16→17→20→18→19) |
| 3 | (3→8→2→1→5→7→6→4→9→11→13→10→12→14→15→16→20→19→18→17) |
| 4 | (4→7→8→2→1→6→5→3→11→10→12→13→14→9→17→19→18→15→20→16) |
| 5 | (2→6→4→8→1→7→5→3→10→13→9→11→14→12→17→19→18→16→20→15) |

**Language Model Tasks.** We evaluate SAIM in language domains using two benchmark suites:

**MergeBench** (He et al., 2025): This benchmark consists of five distinct domains with various tasks in each domain. We use the official Llama-3.2-3B and Llama-3.1-8B model weights in our experiments. Table 7 details the datasets and evaluation metrics for each domain.

**TRACEBench** (Wang et al., 2023): This benchmark evaluates continual learning through a sequence of seven tasks, with each task containing 5,000 samples. We use llama-3.2-3B-Instruct and

Table 7: Datasets used for model evaluation in MergeBench.

| Category | Dataset | Metric | # Data |
|---|---|---|---|
| Instruction-following | IFEval | Prompt level accuracy | 541 |
| Mathematics | GSM8k | EM (8-shot CoT) | 1320 |
| | MATH | EM (0-shot CoT) | 5000 |
| Multilingual understanding | M_MMLU | Accuracy | 60K |
| | M_ARC | Normalized accuracy | 10.34K |
| | M_Hellaswag | Normalized accuracy | 37.35K |
| Coding | Humaneval+ | Pass@1 | 164 |
| | MBPP+ | Pass@1 | 378 |
| Safety | WildGuardTest | RTA | 1730 |
| | HarmBench | RTA | 410 |
| | DoAnythingNow | RTA | 15.14K |
| | XSTest | Accuracy | 450 |

llama-3.2-1B-Instruct as the base model and follow the official task sequence. The evaluation metrics for each task are summarized in Table 8.

Table 8: TRACEBench tasks and evaluation metrics.

| Category | Task | Description | Metric | Avg. Len. |
|---|---|---|---|---|
| Domain-specific | ScienceQA | Science reasoning | Accuracy | 210 |
| | FOMC | Meeting summarization | Accuracy | 51 |
| | MeetingBank | Meeting QA | ROUGE-L | 2853 |
| Multi-lingual | C-STANCE | Stance detection | Accuracy | 127 |
| | 20Minuten | News summarization | SARI | 382 |
| Mathematical reasoning | NumGLUE-cm | Commonsense math | Accuracy | 32 |
| | NumGLUE-ds | Data science reasoning | Accuracy | 21 |

**Evaluation Protocol.** We evaluate our method in two fine-tuning settings:

- **Independent fine-tuning**: Each task is fine-tuned separately from the same pre-trained model, then merged.
- **Sequential fine-tuning**: Each task is fine-tuned from the previous merged model, forming a true continual learning pipeline.

For vision tasks, we follow the FusionBench protocol proposed by Tang et al. (2024); Qiu et al. (2025), reporting average accuracy (ACC) and backward transfer (BWT) as defined in the main text. For MergeBench, each domain score is calculated as the average of the main metrics of all tasks within that domain (e.g., prompt-level accuracy, exact match, pass@1, etc.), and the overall score is the arithmetic mean of the five domain scores. For TRACEBench, the main metric of each of the seven tasks is extracted, and the final score is the arithmetic mean of these seven metrics.

## C.2 IMPLEMENTATION DETAILS

All experiments were conducted on NVIDIA RTX A800(80G) GPUs. Mixed precision (FP16) training was enabled for all experiments to improve efficiency and reduce memory usage. Adam optimizer was adopted with cosine learning rate scheduling and a weight decay of 0.01.

**Experiments on Vision Tasks.** For vision tasks, we used CLIP-ViT models (ViT-B/32, ViT-B/16, ViT-L/14) as the backbone architectures with input resolution 224, following the FusionBench protocol (Tang et al., 2024; Qiu et al., 2025). The global batch size was 128 (32 for ViT-L/14), with gradient accumulation to ensure consistent effective batch size. Each ViT model was fine-tuned on

each task using the Adam optimizer with a learning rate of 1e-5. The batch size was set to 32 for ViT-L-14 and 128 for the other models, with gradient accumulation steps of 4 for ViT-L-14 and 1 for the others. The number of training epochs for each task is consistent with the ISO-C protocol (Marczak et al., 2025). For our SAIM method, we set the balance coefficient $\lambda$ to 0, the perturbation magnitude $\rho$ in SA-BCD to 0.05, and the parameter selection ratio $p$ to 0.3. All experiments use the official training splits, and no data from previous tasks is accessed during fine-tuning or merging. To evaluate robustness to task order, we repeat experiments with five different task permutations (random seeds 42 to 46) and report the mean and standard deviation (Tang et al., 2025; Qiu et al., 2025).

**Experiments on Large Language Models.** For large language model experiments, we conducted continual learning on both the MergeBench (He et al., 2025) and TRACEBench (Wang et al., 2023) benchmarks. For MergeBench, we used the official Llama-3.2-3B and Llama-3.1-8B weights provided by huggingface[2], and performed continual model merging across five domains (instruction, mathematics, multilingual, coding, safety) to simulate the continual learning process (He et al., 2025). In SAIM, the perturbation magnitude $\rho$ in SA-BCD was set to 0.05, the parameter selection ratio $p$ to 0.3, the scaling factor $\alpha$ to 1.0 for Llama-3.2-3B and 1.1 for Llama-3.1-8B, and the balance coefficient $\lambda$ to 0.25. For TRACEBench, we followed the TRACE and MoFO protocols (Chen et al., 2024; Wang et al., 2023), using llama-3.2-3B-Instruct and llama-3.2-1B-Instruct as the base models and performing continual fine-tuning and merging according to the official task sequence (C-STANCE, FOMC, MeetingBank, ScienceQA, NumGLUE-cm, NumGLUE-ds, 20Minuten). Fine-tuning was performed with a learning rate of 2e-5, batch size of 8, and cosine decay schedule. In SAIM, the balance coefficient $\lambda$ was set to 0, with other hyperparameters the same as above, and the sequence length was set to 2048.

## C.3 BASELINE METHODS

All baseline methods evaluated in this work are implemented following their original principles and code logic, with hyperparameters set according to the recommendations in their respective papers. Let $\theta_0$ denote the pre-trained parameters, $\theta_t$ the expert model fine-tuned for task $t$, and the task vector $\boldsymbol{\tau}_t = \theta_t - \theta_0$. After $t$ tasks, the merged model is denoted as $\theta_{\text{merged}}^{(t)}$.

**Continual Fine-tuning (C. Fine-Tuned).** The model is sequentially fine-tuned on each new task dataset $D_t$, updating parameters as $\theta^{(t)} = \text{FineTune}(\theta^{(t-1)}, D_t)$, with $\theta^{(0)} = \theta_0$. This approach adapts the model to each new task but may lead to forgetting previous knowledge.

**Stochastic Weight Averaging (SWA).** SWA incrementally averages the parameters of all task-specific models(Izmailov et al., 2018). At each step, the merged model is updated as $\theta_{\text{SWA}}^{(t)} = \frac{t-1}{t} \theta_{\text{SWA}}^{(t-1)} + \frac{1}{t} \theta_t$, which is equivalent to $\theta_{\text{SWA}}^{(T)} = \theta_0 + \frac{1}{T} \sum_{t=1}^{T} \boldsymbol{\tau}_t$. This method smooths out individual task updates and can improve robustness.

**Continual Task Arithmetic (C. TA).** For each task $t$, the task vector is defined as $\boldsymbol{\tau}_t = \theta_t - \theta_0$. The cumulative task vector up to step $t$ is $\boldsymbol{\tau}_{\text{cum}}^{(t)} = \sum_{i=1}^{t} \boldsymbol{\tau}_i$. The merged model is constructed by applying a scaling factor $\alpha$ to the cumulative task vector, i.e., $\theta_{\text{merged}}^{(t)} = \theta_0 + \alpha \boldsymbol{\tau}_{\text{cum}}^{(t)}$. This approach integrates knowledge from all tasks by accumulating their parameter changes(Ilharco et al., 2022).

**Continual TIES-Merging (C. TIES).** Each task vector $\boldsymbol{\tau}_t$ is pruned by keeping only the top-$k\%$ entries with the largest absolute values, setting the rest to zero. The pruned vector is denoted as $\text{Trim}_k(\boldsymbol{\tau}_t)$. The cumulative pruned vector is updated recursively as $\Delta_{\text{cum}}^{(t)} = \Delta_{\text{cum}}^{(t-1)} + \text{Trim}_k(\boldsymbol{\tau}_t)$, starting from $\Delta_{\text{cum}}^{(0)} = 0$. The merged model is then computed as $\theta_{\text{merged}}^{(t)} = \theta_0 + \alpha \Delta_{\text{cum}}^{(t)}$, where $\alpha$ is a scaling factor. This method emphasizes the most significant parameter changes and reduces interference(Yadav et al., 2023).

**Maximum Magnitude Selection (MagMax).** For each parameter dimension $j$, MagMax selects the update with the largest absolute value among all tasks, i.e., $\Delta\theta_{t,j}^{\text{MagMax}} = \arg\max_{s \leq t} |\boldsymbol{\tau}_{s,j}|$. The

---

[2]https://huggingface.co/MergeBench

merged model is given by $\theta_{\text{merged}}^{(t)} = \theta_0 + \alpha \Delta \theta_t^{\text{MagMax}}$, where $\alpha$ is a scaling coefficient. This approach aims to maximize stability by always choosing the strongest signal for each parameter(Marczak et al., 2024).

**Orthogonal Projection-based Continual Merging (OPCM).** OPCM is a projection-based scheme to mitigate task interference by enforcing orthogonality between parameter updates. Specifically, at each step, the update $\Delta \theta_t$ is projected onto the orthogonal complement of the subspace spanned by previous updates. The merged model is computed as $\theta_t^{\text{merged}} = \theta_0 + \frac{1}{\lambda_t} \left[ \lambda_{t-1} \Delta \theta_{t-1}^{\text{merged}} + \mathcal{P}^{(t-1)}(\Delta \theta_t) \right]$, where $\mathcal{P}^{(t-1)}$ denotes the orthogonal projection operator and $\lambda_t$ is a normalization factor. This approach reduces overlap between tasks and preserves unique information in the merged model(Tang et al., 2025).

**Elastic Weight Consolidation (EWC).** EWC protects past knowledge by penalizing parameter drift weighted by diagonal Fisher information(Kirkpatrick et al., 2017). The objective is to minimize $\mathcal{L}_t(\theta) + \frac{\lambda}{2} \sum_i F_i (\theta_i - \theta_i^\star)^2$, where $\theta^\star$ is the previous optimum and $\lambda$ controls the trade-off between stability and plasticity. This regularization helps retain important parameters for previous tasks.

**Experience Replay (ER).** Current data $D_t$ is mixed with a replay buffer $\mathcal{B}$ of past samples. The optimization objective is $\min_\theta \mathbb{E}_{(x,y) \sim D_t} \mathcal{L}(\theta; x, y) + \mu \mathbb{E}_{(x,y) \sim \mathcal{B}} \mathcal{L}(\theta; x, y)$, with $\mu > 0$ controlling the replay strength. This method alleviates forgetting by revisiting previous data during training(Buzzega et al., 2020).

## C.4 PARAMETER CHANGE ANALYSIS SETUP

In our parameter change analysis experiments, we used the ViT-B/32 model and analyzed parameter changes after fine-tuning on five categories of visual tasks, each containing three specific datasets:

- OCR (Optical Character Recognition): MNIST, EMNIST, KMNIST
- VQA (Visual Question Answering): STL10, CIFAR100, RenderedSST2
- Geometry (Geometric Recognition): GTSRB, RESISC45, EuroSAT
- Chart (Chart and Scene Classification): CIFAR10, Food101, SUN397
- Grounding (Object Grounding and Segmentation): Cars, OxfordIIITPet, DTD

For each task category, we performed fine-tuning using both the standard Adam optimizer and our proposed SA-BCD optimizer, with a parameter selection ratio $p = 0.3$ and perturbation magnitude $\rho = 0.05$. All fine-tuning was conducted with a learning rate of $1\text{e}-5$ and weight decay of $0.01$. To ensure reproducibility, we set the random seed to 42 and fine-tuned separately on each of the three datasets for every task category. After fine-tuning, we computed the absolute difference between the fine-tuned and pre-trained model parameters, took the logarithm ($\log_{10}$), and plotted histograms to visualize the distribution of parameter changes under different optimizers.

## C.5 WEIGHT DISENTANGLEMENT VISUALIZATION SETUP

To evaluate the effectiveness of SA-BCD fine-tuning in reducing parameter interference and improving weight disentanglement, we conduct visualization experiments under two settings: (1) merging two task-specific models across two tasks, and (2) merging all eight task-specific models across two tasks.

In the first setting, the merged model is parameterized as $\theta_{\text{merge}} = \theta_0 + \alpha_1 \tau_1 + \alpha_2 \tau_2$, where $\tau_1$ and $\tau_2$ are the task vectors obtained by fine-tuning on tasks 1 and 2, respectively. In the second setting, the merged model is parameterized as $\theta_{\text{merge}} = \theta_0 + \alpha_1 \tau_1 + \alpha_2 \tau_2 + \sum_{s \notin \{1,2\}} \alpha_s \tau_s$, where $\tau_s$ denotes the task vectors for the remaining six tasks and $\alpha_s$ is obtained through grid search to maximize the average accuracy.

For both settings, we vary $(\alpha_1, \alpha_2)$ over a grid from $-0.5$ to $1.5$ with 21 evenly spaced points along each axis, resulting in a $21 \times 21$ grid. For each $(\alpha_1, \alpha_2)$ pair, we compute the disentanglement error $\xi(\alpha_1, \alpha_2)$ using the merged model obtained via SA-BCD fine-tuning. The error values are visualized

using contour plots, which highlight regions in the parameter space where weight disentanglement is stronger. Since the task coefficients are real-valued, contour plots provide an effective way to illustrate the variations in loss landscape and disentanglement error across the continuous $(\alpha_1, \alpha_2)$ space.

# D  ADDITIONAL RESULTS

This section presents supplementary experimental results to further demonstrate the effectiveness of SAIM. We include detailed task performance (D.1), accuracy matrices across task sequences (D.2), multi-task weight disentanglement and subspace alignment analysis (D.3), ablation studies (D.4), parameter sensitivity analysis (D.5), supplementary analysis of MergeBench merging results including the Math task (D.6), and additional evaluation on TRACEBench with Llama-3.2-1B (D.7).

Table 9: Test set accuracy comparisons on different downstream tasks.

| Model | SUN397 | Cars | RESISC45 | EuroSAT | SVHN | GTSRB | MNIST | DTD | Flowers102 | PCAM |
|---|---|---|---|---|---|---|---|---|---|---|
| **ViT-B/32** | | | | | | | | | | |
| C. FINE-TUNED | 53.9 | 38.2 | 64.7 | **98.7** | 45.4 | 34.4 | 86.7 | 58.4 | 57.5 | 67.7 |
| AVERAGE (SWA) | 64.2 | 59.6 | 64.8 | 60.9 | 47.3 | 43.1 | 71.8 | 46.4 | 66.5 | 63.9 |
| C. TA | 62.0 | 53.7 | 60.9 | 58.1 | 48.5 | 48.9 | 79.4 | 46.1 | 61.1 | 73.4 |
| C. TIES | 62.5 | 49.1 | 55.8 | 50.9 | 54.6 | 49.3 | 82.0 | 46.7 | 58.5 | 69.9 |
| MAGMAX-IND | 63.6 | 53.1 | 59.7 | 49.1 | 53.8 | 53.1 | 79.8 | 43.2 | 56.9 | 75.1 |
| OPCM | 64.4 | 51.1 | 66.0 | 71.7 | **66.1** | 56.0 | 90.2 | 40.4 | 64.9 | 80.2 |
| **SAIM (Ours)** | **68.5** | **60.3** | **84.5** | 74.9 | 65.3 | **76.1** | 86.0 | **66.2** | **68.2** | **84.9** |
| **ViT-B/16** | | | | | | | | | | |
| C. FINE-TUNED | 62.7 | 58.0 | 67.6 | **99.1** | 46.0 | 29.2 | 93.9 | 61.9 | 64.1 | 75.2 |
| AVERAGE (SWA) | 67.1 | 64.6 | 69.3 | 63.4 | 62.4 | 52.7 | 80.7 | 46.6 | 71.8 | 63.1 |
| C. TA | 65.8 | 57.5 | 63.8 | 59.5 | 64.7 | 54.0 | 88.0 | 45.3 | 67.5 | 67.1 |
| C. TIES | 64.2 | 52.9 | 60.9 | 53.0 | 62.8 | 48.8 | 88.4 | 45.0 | 61.3 | 68.5 |
| MAGMAX-IND | 65.8 | 51.8 | 57.8 | 42.6 | 54.4 | 43.7 | 83.0 | 42.8 | 60.4 | 69.8 |
| OPCM | 67.9 | 55.9 | 73.7 | 77.5 | **74.4** | 63.2 | 94.1 | 49.2 | **72.3** | 79.6 |
| **SAIM (Ours)** | **71.9** | **71.1** | **86.5** | 85.0 | 70.4 | **84.3** | 92.3 | **70.9** | 70.8 | **80.5** |
| **ViT-L/14** | | | | | | | | | | |
| C. FINE-TUNED | 69.5 | 73.6 | 78.3 | **99.2** | 59.3 | 49.3 | **98.6** | 69.7 | 83.2 | 78.3 |
| AVERAGE (SWA) | 70.7 | 77.7 | 76.4 | 75.3 | 69.5 | 62.1 | 93.7 | 57.7 | 80.0 | 73.6 |
| C. TA | 70.4 | 74.1 | 73.9 | 66.3 | 69.9 | 65.6 | 95.1 | 56.6 | 78.6 | 70.4 |
| C. TIES | 69.7 | 70.3 | 65.3 | 47.9 | 76.1 | 63.6 | 94.7 | 54.4 | 77.9 | 72.3 |
| MAGMAX-IND | 73.1 | 73.7 | 75.6 | 64.6 | 73.7 | 68.8 | 94.6 | 56.1 | 78.0 | 71.7 |
| OPCM | 73.1 | 78.3 | 82.4 | 80.2 | 80.8 | 80.4 | 97.4 | 61.6 | **84.8** | 76.3 |
| **SAIM (Ours)** | **77.8** | **85.8** | **93.9** | 92.6 | **86.2** | **91.3** | 97.3 | **78.1** | 82.9 | **90.3** |

| Model | FER2013 | OxfordIIITPet | STL10 | CIFAR100 | CIFAR10 | Food101 | FashionMNIST | EMNIST | KMNIST | RenderedSST2 |
|---|---|---|---|---|---|---|---|---|---|---|
| **ViT-B/32** | | | | | | | | | | |
| C. FINE-TUNED | **58.3** | 68.5 | 86.7 | 40.2 | 70.5 | 50.0 | **90.7** | 72.4 | **54.5** | 54.5 |
| AVERAGE (SWA) | 50.2 | **84.1** | **97.0** | **69.8** | 92.7 | **80.4** | 71.3 | 15.0 | 11.5 | 61.8 |
| C. TA | 51.4 | 82.3 | 94.9 | 64.6 | 91.4 | 71.9 | 73.9 | 17.8 | 12.2 | 59.9 |
| C. TIES | 49.5 | 81.3 | 95.2 | 63.7 | 91.2 | 70.2 | 73.7 | 17.8 | 16.9 | 59.8 |
| MAGMAX-IND | 56.5 | 79.9 | 94.6 | 68.7 | 91.9 | 73.8 | 74.3 | 18.3 | 15.4 | 63.9 |
| OPCM | 55.8 | 82.9 | 95.9 | 67.6 | **92.8** | 74.0 | 76.3 | 22.4 | 18.3 | 64.6 |
| **SAIM (Ours)** | 49.9 | **84.1** | 95.5 | 69.6 | 92.5 | 78.2 | 81.2 | **74.3** | 42.3 | **69.3** |
| **ViT-B/16** | | | | | | | | | | |
| C. FINE-TUNED | **60.5** | 84.5 | 90.5 | 38.8 | 73.6 | 61.9 | **89.7** | 83.3 | **51.5** | 72.8 |
| AVERAGE (SWA) | 50.9 | 89.6 | **98.0** | 72.9 | 94.2 | **85.9** | 73.3 | 15.6 | 12.4 | 62.5 |
| C. TA | 50.7 | 89.3 | 97.0 | 68.0 | 93.1 | 80.3 | 75.7 | 18.1 | 16.7 | 61.8 |
| C. TIES | 50.4 | 87.9 | 96.3 | 63.1 | 91.7 | 78.0 | 75.0 | 23.4 | 24.9 | 61.5 |
| MAGMAX-IND | 57.7 | 88.8 | 97.5 | 71.5 | 94.4 | 81.3 | 77.2 | 24.5 | 25.0 | 59.4 |
| OPCM | 59.5 | **91.8** | 97.7 | 73.2 | 94.7 | 83.1 | 81.3 | 26.5 | 23.4 | 66.8 |
| **SAIM (Ours)** | 49.7 | 90.1 | 97.8 | **76.1** | 95.0 | 85.3 | 86.2 | **92.9** | 49.9 | **74.2** |
| **ViT-L/14** | | | | | | | | | | |
| C. FINE-TUNED | **68.0** | 92.1 | 94.5 | 60.5 | 85.7 | 74.8 | **93.1** | 89.0 | 59.2 | 78.8 |
| AVERAGE (SWA) | 52.7 | 94.2 | 99.2 | 81.7 | 97.0 | 90.7 | 77.4 | 16.1 | 10.4 | 66.1 |
| C. TA | 55.7 | 94.2 | 98.6 | 79.1 | 91.6 | 87.6 | 80.8 | 17.6 | 10.6 | 63.6 |
| C. TIES | 57.6 | 93.5 | 97.8 | 74.0 | 95.6 | 84.7 | 79.7 | 20.2 | 12.6 | 58.4 |
| MAGMAX-IND | 52.9 | 93.9 | 98.7 | 82.1 | 97.3 | 89.5 | 81.6 | 19.2 | 11.1 | 68.4 |
| OPCM | 61.8 | **95.4** | 99.2 | 83.0 | 97.8 | **90.9** | 86.0 | 26.4 | 14.7 | 71.0 |
| **SAIM (Ours)** | 53.5 | 95.2 | **99.6** | **85.4** | **98.3** | 90.2 | 90.7 | **95.4** | **82.4** | **82.5** |

## D.1  DETAILED TASK PERFORMANCE

Table 9 extends the results from Table 1 in the main text, providing a detailed breakdown of accuracy for each of the 20 visual tasks after merging. We compare various methods (SWA, Task Arithmetic,

TIES-Merging, MagMAX-IND, OPCM, and our proposed SAIM) across three CLIP-ViT backbones (ViT-B/32, ViT-B/16, ViT-L/14).

As shown in Table 9, SAIM achieves the best performance on most tasks. Particularly significant improvements are observed on complex tasks such as SUN397, RESISC45, and GTSRB, where SAIM outperforms the second-best method by approximately 5-20 percentage points. This advantage becomes more pronounced with larger models (ViT-L/14), indicating that SAIM effectively leverages the expressive capacity of larger architectures. Notably, on certain tasks like FER2013, other methods may show slight advantages, suggesting that task characteristics can influence the optimal merging strategy.

Moreover, we observe that all methods face challenges on certain tasks (e.g., EMNIST and KM-NIST), likely due to their significant distribution shift from pre-training data. Even on these challenging tasks, however, SAIM demonstrates superior generalization capabilities, validating our method's effectiveness in cross-task knowledge transfer.

## D.2 Accuracy Matrices Across Task Sequences

Figure 4 shows the accuracy matrices of SAIM under different model sizes and task numbers, helping visualize how well knowledge is retained during continual learning. In each matrix, rows indicate the order in which tasks are merged, and columns show the test accuracy for each task. Diagonal entries represent the accuracy on the current task, while off-diagonal entries reflect how well the model remembers previous tasks and transfers knowledge.

From these matrices, we observe several clear trends. First, as the model size increases from ViT-B/32 to ViT-L/14, the overall accuracy improves and the values become more uniformly high (see subfigures a-c). Second, when the number of tasks increases from 8 to 20, SAIM remains stable, keeping high accuracy even for earlier tasks in longer sequences (see subfigures d-f), with most values above 80% in the ViT-L/14 20-task matrix.

SAIM also performs well when switching between very different tasks. For example, in the ViT-L/14 results, the model maintains high accuracy when moving from scene recognition (SUN397) to digit recognition (MNIST) and then to remote sensing (RESISC45). The consistently bright colors in the matrices indicate that SAIM can integrate knowledge from diverse domains with little interference between tasks.

## D.3 Multi-task Weight Disentanglement and Subspace Alignment Analysis

**Multi-task Weight Disentanglement Analysis.** To extend our analysis beyond the two-task scenario presented in the main text, we evaluate weight disentanglement in complex multi-task settings. For multi-task scenarios, the disentanglement error is defined as:

$$\xi_{\text{all}}(\alpha_1, \alpha_2) = \sum_{t=1}^{2} \mathbb{E}_{\boldsymbol{x} \in X^{(t)}} \left[ \text{dist}\big(f(\boldsymbol{x}; \boldsymbol{\theta}_t^{ref}),\ f(\boldsymbol{x}; \boldsymbol{\theta}_{merged})\big) \right] \tag{35}$$

where $\boldsymbol{\theta}_{merged} = \boldsymbol{\theta}_0 + \alpha_1 \boldsymbol{\tau}_1 + \alpha_2 \boldsymbol{\tau}_2 + \sum_{s \notin \{1,2\}} \alpha_s \boldsymbol{\tau}_s$ and $\boldsymbol{\theta}_t^{ref} = \boldsymbol{\theta}_0 + \alpha_t \boldsymbol{\tau}_t + \sum_{s \notin \{1,2\}} \alpha_s \boldsymbol{\tau}_s$.

The visualization of $\xi_{\text{all}}$ under different $\alpha_1$ and $\alpha_2$ (see Fig. 5) shows that the merged model under SA-BCD fine-tuning achieves lower disentanglement errors in the multi-task scenario, further confirming that our method enables the merged model to better reflect the independent contributions of each task even in more complex settings.

**Subspace Alignment with Increasing Number of Tasks.** ISO-C (Marczak et al., 2025) established that the subspace alignment ratio (SAR) between the merged model and each task model correlates strongly with merging performance. SAR quantifies the overlap between the task-specific parameter update matrix and the principal subspace of the merged model. We plot the curves of average SAR and test accuracy as the number of tasks increases (see Fig. 6). The results show that the SAIM method generally maintains higher SAR and accuracy compared to baselines as the number of tasks increases, indicating better subspace alignment between the merged model and each task model, thereby enhancing multi-task knowledge integration and overall performance.

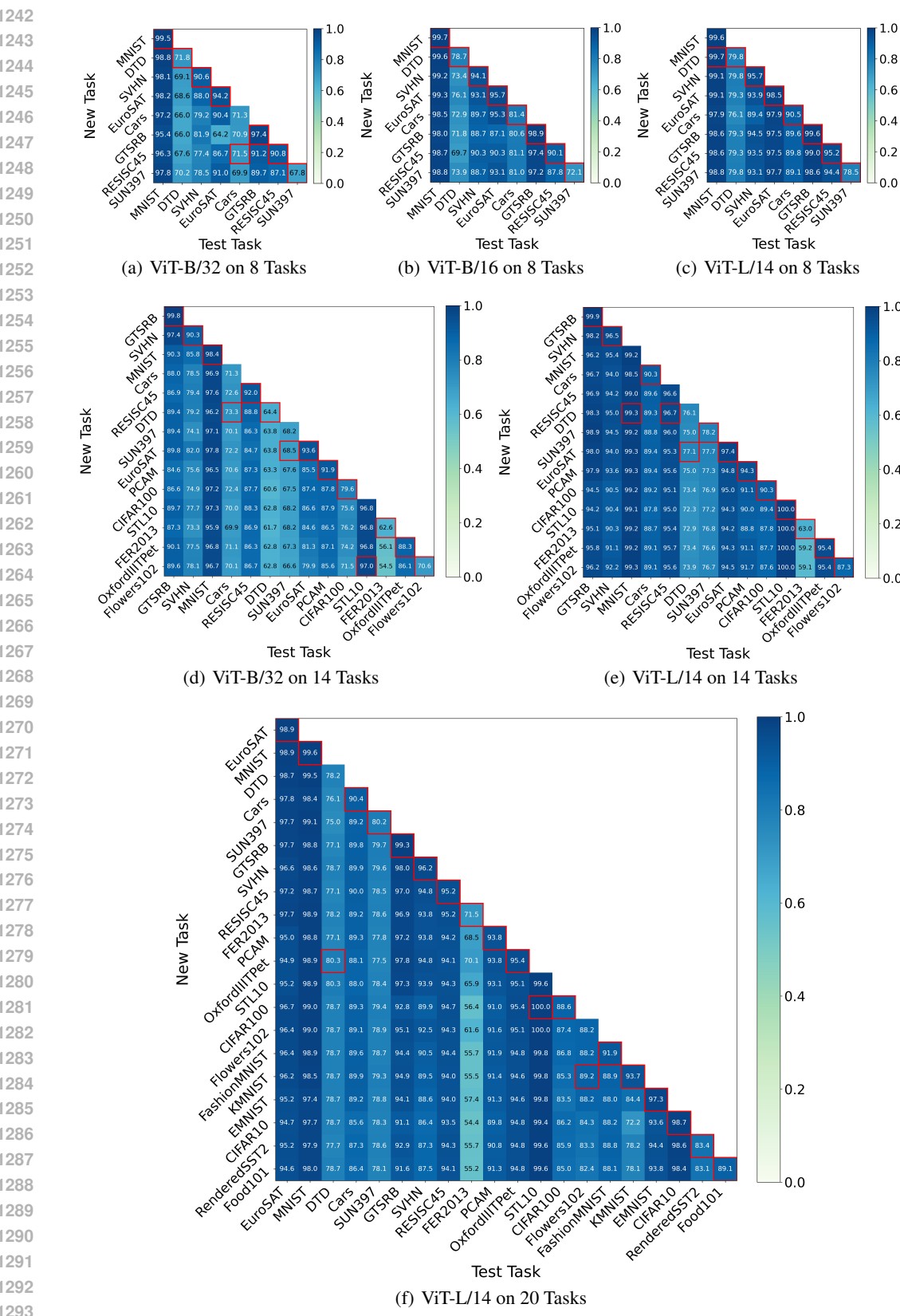

Figure 4: Accuracy matrices of SAIM for different model scales and task settings: (a)-(c) show 8 tasks, (d)-(e) 14 tasks, and (f) 20 tasks. Rows indicate new tasks, columns indicate test tasks.

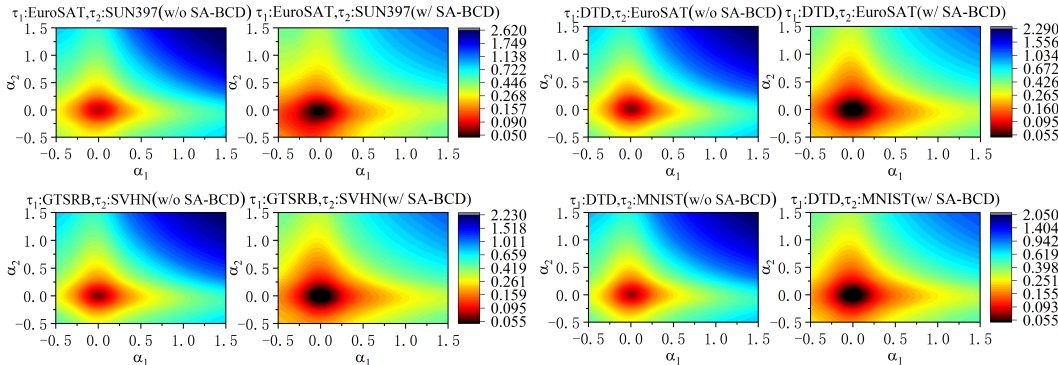

Figure 5: Visualization of weight disentanglement error in the eight-task merging scenario. Only two task coefficients are adjusted while the others are fixed, demonstrating the weight disentanglement capability of the merged model in a multi-task setting.

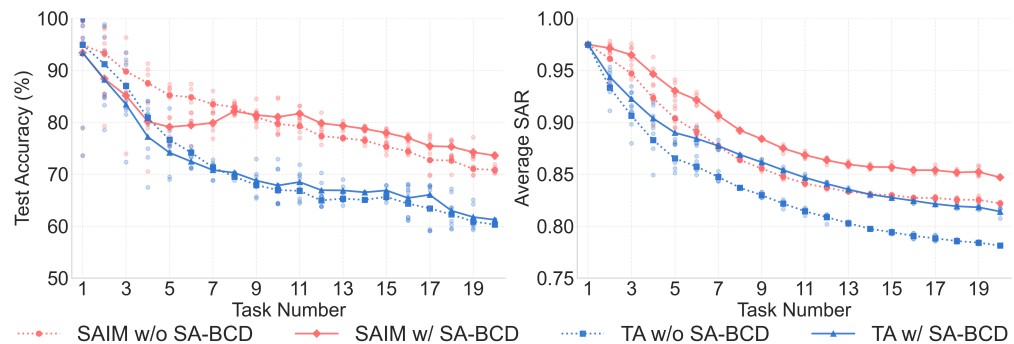

Figure 6: Average subspace alignment ratio (SAR) and test accuracy versus the number of tasks on ViT-B/32. Results are averaged over five different task orders. The left panel shows the evolution of test accuracy during continual learning for different methods, while the right panel shows the change in average SAR. The SAIM method generally achieves higher accuracy and SAR as the number of tasks increases, indicating enhanced multi-task knowledge integration and subspace alignment.

## D.4 ABLATION STUDIES AND ANALYSIS

To further validate the effectiveness and generalizability of our method, we conducted two sets of ablation experiments. The experimental settings are consistent with those described in Section 4.1, including hyperparameters such as learning rate, perturbation magnitude $\rho$, and parameter selection ratio $p$. All experiments were repeated under five different task orders, and we report the mean and standard deviation.

**Impact of SA-BCD Fine-tuning on Model Merging Methods.** We compared the performance of mainstream model merging methods under standard Adam and SA-BCD fine-tuning (see Table 10 and Table 11). The results show that SA-BCD fine-tuning significantly improves the accuracy of most methods, especially Task Arithmetic and TIES-Merging, with the highest improvement reaching 3.5%. The SAIM method also achieves consistent gains, indicating that sharpness-aware fine-tuning can be effectively combined with existing merging strategies to further enhance continual learning performance. It is worth noting that SWA shows a slight decrease under SA-BCD, which may be due to incompatibility between its parameter averaging strategy and the sparsification and sharpness optimization introduced by SA-BCD. These results demonstrate the general applicability of SA-BCD.

**Effect of Independent vs. Sequential Fine-tuning.** We further compared the impact of independent fine-tuning (each task starts from the pre-trained model) and sequential fine-tuning (each task starts from the previously merged model) on merging performance (see Table 12). The results demonstrate that sequential fine-tuning significantly improves the accuracy of all methods,

Table 10: Impact of SA-BCD Fine-tuning on Different Model Merging Methods (20 tasks, ViT-B/32).

| Method | Accuracy (%) | | Improvement |
|---|---|---|---|
| | w/o SA-BCD | w/ SA-BCD | |
| Average (SWA) | $61.1_{\pm 0.0}$ | $59.6_{\pm 0.0}$ | -1.5 |
| C. Task Arithmetic | $60.0_{\pm 0.0}$ | $61.2_{\pm 0.1}$ | +1.2 |
| C. TIES-Merging | $59.9_{\pm 0.7}$ | $63.4_{\pm 0.5}$ | +3.5 |
| **SAIM (Ours)** | $\mathbf{70.8}_{\pm 0.6}$ | $\mathbf{73.6}_{\pm 0.3}$ | +2.8 |

Table 11: Impact of SA-BCD Fine-tuning on Different Model Merging Methods (20 tasks, ViT-B/16).

| Method | Accuracy (%) | | Improvement |
|---|---|---|---|
| | w/o SA-BCD | w/ SA-BCD | |
| Average (SWA) | $64.8_{\pm 0.0}$ | $62.4_{\pm 0.0}$ | -2.4 |
| C. Task Arithmetic | $64.2_{\pm 0.0}$ | $65.6_{\pm 0.0}$ | +1.4 |
| C. TIES-Merging | $63.0_{\pm 1.6}$ | $66.5_{\pm 0.7}$ | +3.5 |
| **SAIM (Ours)** | $\mathbf{77.0}_{\pm 0.5}$ | $\mathbf{79.0}_{\pm 0.9}$ | +2.0 |

with gains exceeding 10% in some cases. This phenomenon may be attributed to two factors: (1) sequential fine-tuning encourages more consistent update directions across tasks, reducing sign conflicts and parameter interference during merging (Marczak et al., 2024); (2) knowledge transfer between tasks is more sufficient, enabling the merged model to better accumulate and retain multi-task information. SAIM consistently achieves the highest accuracy under both fine-tuning strategies, highlighting its strong robustness and scalability.

Table 12: Comparison of independent (Ind) and sequential (Seq) fine-tuning for SAIM, Task Arithmetic, and TIES-Merging on ViT-B/32 and ViT-B/16.

| Method | FT | ViT-B/32 | | | ViT-B/16 | | |
|---|---|---|---|---|---|---|---|
| | | 8 tasks | 14 tasks | 20 tasks | 8 tasks | 14 tasks | 20 tasks |
| C. Task Arithmetic | Ind | $67.5_{\pm 0.0}$ | $66.5_{\pm 0.0}$ | $60.0_{\pm 0.0}$ | $77.1_{\pm 0.0}$ | $70.9_{\pm 0.0}$ | $64.2_{\pm 0.0}$ |
| | Seq | $77.8_{\pm 0.7}$ | $75.9_{\pm 0.6}$ | $71.5_{\pm 1.0}$ | $81.0_{\pm 0.3}$ | $78.2_{\pm 0.7}$ | $76.3_{\pm 0.6}$ |
| C. TIES-Merging | Ind | $49.0_{\pm 10.2}$ | $66.2_{\pm 0.6}$ | $59.9_{\pm 0.7}$ | $66.8_{\pm 3.7}$ | $70.5_{\pm 0.8}$ | $63.0_{\pm 1.6}$ |
| | Seq | $78.6_{\pm 0.6}$ | $74.9_{\pm 0.9}$ | $71.8_{\pm 0.5}$ | $81.7_{\pm 0.9}$ | $79.1_{\pm 0.6}$ | $77.1_{\pm 0.7}$ |
| **SAIM (Ours)** | Ind | $82.1_{\pm 0.7}$ | $\mathbf{78.7}_{\pm 0.5}$ | $73.6_{\pm 0.3}$ | $86.2_{\pm 0.5}$ | $82.0_{\pm 0.5}$ | $79.0_{\pm 0.9}$ |
| | Seq | $\mathbf{82.5}_{\pm 1.4}$ | $\mathbf{78.7}_{\pm 1.3}$ | $\mathbf{76.4}_{\pm 2.0}$ | $\mathbf{87.9}_{\pm 0.6}$ | $\mathbf{83.9}_{\pm 1.0}$ | $\mathbf{81.9}_{\pm 1.0}$ |

## D.5 PARAMETER SENSITIVITY ANALYSIS

To comprehensively evaluate the robustness of SAIM and determine optimal hyperparameter configurations, we systematically analyze three key hyperparameters: the balance factor $\lambda$, parameter selection ratio $p$, and perturbation magnitude $\rho$. All experiments are conducted on the ViT-B/32 backbone, considering three settings with 8, 14, and 20 tasks. For each setting, we repeat experiments with five different task orders to ensure reliability and stability.

**Balance Factor $\lambda$ Analysis:** As shown in Figure 7(a), we perform a detailed grid search for $\lambda$ in the range $[-0.5, 0.5]$. The balance factor $\lambda$ controls the trade-off between historical knowledge and new task updates, which is crucial for continual learning. Results indicate that for all task numbers, the model achieves peak average accuracy when $\lambda$ is near zero. Specifically, accuracy reaches approximately 83% for 8 tasks, 78% for 14 tasks, and 73% for 20 tasks when $\lambda \approx 0$. Performance drops significantly as $\lambda$ deviates towards either extreme, highlighting the importance of balancing historical and new knowledge. Notably, the optimal $\lambda$ remains stable near zero as the number of tasks increases, demonstrating good cross-task robustness.

**Parameter Selection Ratio $p$ Analysis:** We search $p$ in the range $[0.1, 1.0]$, which determines the proportion of parameters updated in each SA-BCD iteration. As shown in Figure 7(b), the model achieves optimal performance when $p \approx 0.3$. Smaller $p$ values (e.g., 0.1) result in insufficient updates, limiting adaptation to new tasks, while larger $p$ values (e.g., 0.8 or 1.0) cause excessive parameter changes, disrupting existing knowledge and increasing task interference. Especially as the number of tasks grows, moderate sparsity ($p \approx 0.3$) becomes more important for reducing interference and maintaining generalization. This finding aligns with prior works such as MoFO and DARE, which emphasize the critical role of update sparsity in mitigating catastrophic forgetting.

**Perturbation Magnitude $\rho$ Analysis:** We investigate the effect of $\rho$ in $[0, 0.005, 0.05, 0.1, 0.5, 1.5]$, which controls the gradient perturbation size in SA-BCD. As shown in Figure 7(c), moderate per-

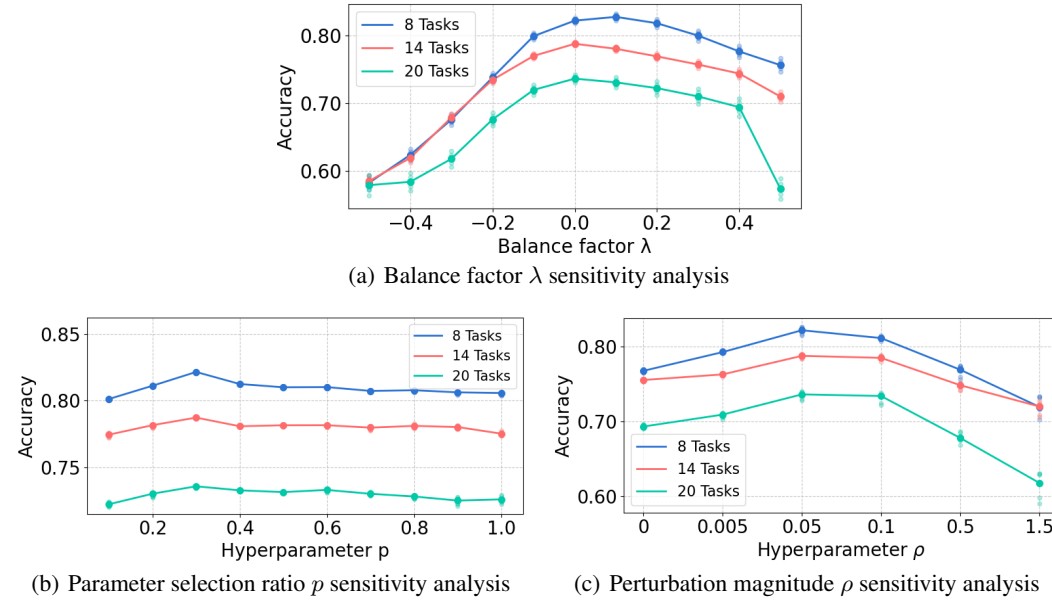

(a) Balance factor $\lambda$ sensitivity analysis

(b) Parameter selection ratio $p$ sensitivity analysis

(c) Perturbation magnitude $\rho$ sensitivity analysis

Figure 7: Parameter sensitivity analysis for SAIM on ViT-B/32 with 8, 14, and 20 tasks: (a) balance factor $\lambda$, (b) parameter selection ratio $p$, and (c) perturbation magnitude $\rho$. All results are averaged over five different task orders.

turbation ($\rho \approx 0.05$) significantly improves model performance. Too small values (e.g., 0 or 0.005) hinder exploration of flat regions in the loss landscape, while too large values (e.g., 0.5) destabilize optimization. The impact of $\rho$ is more pronounced with 20 tasks, indicating that as task complexity increases, guiding the model towards flatter minima becomes increasingly important. This supports our theoretical analysis that flat minima help reduce task interference and enhance knowledge retention in continual learning.

### D.6 RESULTS OF FIVE-TASK MODEL MERGING ON MERGEBENCH WITH LLAMA-3.1-8B

Table 13 presents the complete results of merging all five task models on Llama-3.1-8B. Notably, the Math fine-tuned model shows lower performance than the pre-trained model (0.4026 vs. 0.5625), which stems from a distribution mismatch between the fine-tuning datasets and the gsm8k evaluation set. This phenomenon occurs because MergeBench was designed to evaluate model merging across diverse capability domains, using specialized training data that sometimes differs from evaluation benchmarks to test generalization. Our SAIM approach achieves the highest average score (0.5512), outperforming Task Arithmetic (0.5069) and other baselines.

Table 13: Comparison of Model Merging Methods on MergeBench with Llama-3.1-8B.

| Method | Task Performance (Average Score) | | | | | Average Score |
|---|---|---|---|---|---|---|
| | Instruction | Math | Multilingual | Coding | Safety | |
| Pre-Trained | 0.0795 | 0.5625 | 0.5324 | 0.3631 | 0.3698 | 0.3815 |
| Fine-Tuned | 0.3752 | 0.4026 | 0.5395 | 0.5627 | 0.7741 | 0.5308 |
| SWA | 0.0786 | 0.7263 | **0.5275** | **0.4966** | 0.5731 | 0.4804 |
| TIES-Merging | 0.0619 | 0.7771 | 0.5190 | 0.4653 | 0.6349 | 0.4916 |
| MagMAX | 0.2089 | 0.7847 | 0.4652 | 0.4525 | 0.6119 | 0.5046 |
| DARE | 0.2079 | 0.7430 | 0.4730 | 0.4738 | 0.5944 | 0.4984 |
| Task Arithmetic | 0.1950 | 0.7718 | 0.4892 | 0.4919 | 0.5867 | 0.5069 |
| **SAIM (Ours)** | **0.3262** | **0.7961** | 0.4642 | 0.4753 | **0.6942** | **0.5512** |

### D.7 PERFORMANCE EVALUATION ON TRACEBENCH WITH LLAMA-3.2-1B

We also evaluated our method using the smaller Llama-3.2-1B-Instruct model on TRACEBench. Table 14 presents these results, which demonstrate similar trends to those observed with the 3B model (see Table 4). SAIM achieves the highest average score (0.4776), outperforming other baselines by a significant margin. This consistency across model sizes highlights the scalability and effectiveness of our approach.

Table 14: Comparison of Model Merging Methods on TRACEBench with Llama-3.2-1B-Instruct.

| Method | Task Performance | | | | | | | Average Score |
|---|---|---|---|---|---|---|---|---|
| | C-STANCE | FOMC | MeetingBank | ScienceQA | NumGLUE-cm | NumGLUE-ds | 20Minuten | |
| Pre-Trained | 0.3386 | 0.2581 | 0.2036 | 0.6780 | 0.1220 | 0.1646 | 0.3802 | 0.3064 |
| Fine-Tuned | 0.4980 | 0.5988 | 0.3707 | 0.8524 | 0.3902 | 0.5793 | 0.3880 | 0.5254 |
| DARE | 0.4071 | 0.4073 | 0.2241 | 0.6870 | 0.1707 | 0.3659 | 0.3728 | 0.3764 |
| SWA | 0.4206 | **0.5040** | 0.2179 | 0.7340 | 0.2195 | 0.3171 | 0.3813 | 0.3992 |
| Task Arithmetic | 0.4134 | 0.4335 | 0.2303 | 0.7100 | 0.2439 | 0.3902 | 0.3803 | 0.4002 |
| TIES-Merging | 0.4114 | 0.4254 | 0.2336 | 0.7210 | 0.2683 | 0.3902 | 0.3803 | 0.4043 |
| MagMAX-IND | **0.4555** | 0.4597 | 0.2370 | 0.7130 | 0.2439 | 0.3476 | 0.3848 | 0.4059 |
| **SAIM (Ours)** | 0.4517 | 0.4899 | **0.2454** | **0.7742** | **0.4878** | **0.5061** | 0.3880 | **0.4776** |

## E ALGORITHMIC DETAILS

We present the full pseudocode for the two core components of our SAIM framework: the Sharpness-Aware Block Coordinate Descent (SA-BCD) optimizer (Algorithm 1) and the Sharpness-Aware Isotropic Merging (SAIM) procedure (Algorithm 2).

---

**Algorithm 1** Sharpness-Aware Block Coordinate Descent (SA-BCD) Optimizer Fine-tuning

---

**Input:** Initial model parameters $\Theta_0$, learning rate $\eta$, momentum parameters $\beta_1, \beta_2$, small constant $\epsilon$, perturbation magnitude $\rho$, parameter selection ratio $p$

**Output:** Fine-tuned model parameters $\Theta_T$

1: Initialize momentum estimates $m_0 = 0, v_0 = 0$
2: **for** $t = 1$ to $T$ **do**
3:     // Sample batch data $D_t$ and compute gradient
4:     $g_t = \nabla_\Theta L(\Theta_{t-1}, D_t)$
5:     // Update first-order momentum
6:     $m_t = \beta_1 m_{t-1} + (1 - \beta_1) g_t$
7:     Select index set $\Omega_t$ corresponding to top $p\%$ values of $|m_t|$
8:     // Compute worst-case perturbation on selected subset
9:     $\epsilon_{t\Omega}^* = \rho \frac{g_t(\Omega_t)}{\|g_t(\Omega_t)\|_2}$
10:     // Update second-order momentum
11:     $v_t = \beta_2 v_{t-1} + (1 - \beta_2) g_t^2$
12:     // Bias correction
13:     $\hat{m}_t = \frac{m_t}{1-\beta_1^t}, \hat{v}_t = \frac{v_t}{1-\beta_2^t}$
14:     // Compute gradient at perturbed points
15:     $g_t' = \nabla_\Theta L(\Theta_{t-1} + \epsilon_{t\Omega}^*, D_t)$
16:     // Selective parameter update
17:     **for** each parameter $i$ **do**
18:         **if** $i \in \Omega_t$ **then**
19:             $\Theta_{t,i} = \Theta_{t-1,i} - \eta \frac{\hat{m}_{t,i}}{\sqrt{\hat{v}_{t,i}} + \epsilon} g_{t,i}'$
20:         **else**
21:             $\Theta_{t,i} = \Theta_{t-1,i}$         ▷ remain unchanged
22:         **end if**
23:     **end for**
24: **end for**
25: **return** $\Theta_T$

---

---

**Algorithm 2** Sharpness-Aware Isotropic Merging (SAIM) for Continual Learning

---

**Input:** Pre-trained model parameters $\theta_{pre}$, task sequence $\mathcal{T} = \{T_1, T_2, \ldots, T_N\}$, balance coefficient $\lambda$, scaling factor $\alpha$

**Output:** Final model parameters $\theta_{final}$

1: $\theta_0 \leftarrow \theta_{pre}$
2: **for** $t = 1, 2, 3, \ldots, N$ **do**
3:     // Sharpness-Aware Fine-tuning (see Algorithm 1)
4:     $\theta_{T_t} \leftarrow$ SA-BCD-Finetune$(\theta_{t-1}, T_t, \eta, \ldots)$
5:     $\Delta_{T_t} = \theta_{T_t} - \theta_{t-1}$
6:     $\Delta_{merged} \leftarrow \emptyset$
7:     **for** each parameter layer $k$ **do**
8:         $\Delta_{cum}^k = \theta_{t-1}^k - \theta_{pre}^k$
9:         $\Delta_{com} = (1 + \lambda)\Delta_{cum}^k + (1 - \lambda)\Delta_{T_t}^k$
10:       // Singular value decomposition
11:       $\Delta_{com} = U\Sigma V^\top$
12:       // Adaptive singular value balancing
13:       $\bar{\sigma} = \frac{1}{r}\sum_{i=1}^r \sigma_i$
14:       $\hat{\Sigma} = \bar{\sigma} + (\Sigma - \bar{\sigma}) \times \frac{1}{\sqrt{t}}$
15:       $\Delta_{merged}^k \leftarrow U\hat{\Sigma}V^\top$
16:     **end for**
17:     // Update model parameters
18:     $\theta_t \leftarrow \theta_0 + \alpha\Delta_{merged}$
19: **end for**
20: **return** $\theta_{final} \leftarrow \theta_N$

---

