# OpenReview forum: "Merge to Remember: Sharpness-Aware Isotropic Merging for Continual Learning"
_ICLR.cc/2026/Conference — Submitted to ICLR 2026_

### Official Review · Reviewer_fzut · 2025-10-22

**Soundness:** 3
**Presentation:** 3
**Contribution:** 3
**Rating:** 6
**Confidence:** 4

**Summary:**

The authors investigate continual learning with pre-trained models and argue that catastrophic forgetting and task interference still persist. They analyze forgetting from the perspective of the loss landscape and dominant singular values, revealing how sharp minima contribute to instability across tasks. To address this, they propose a Sharpness-Aware Module that guides the model toward flatter minima while selectively updating task-sensitive parameters. However, the experimental evaluation is somewhat limited — it lacks comparison with more recent vision-language model (VLM)-based approaches, which have become strong baselines in continual learning research.

**Strengths:**

1) Provides a theoretical and empirical linkage between forgetting and sharpness, a relatively under explored perspective in CL.
2) The sharpness-aware optimization idea is conceptually simple yet well-motivated for continual adaptation.
3) Compatible with pre-trained architectures, allowing easy integration into practical pipelines.
4) The singular value analysis offers a useful diagnostic for understanding representational drift and interference

**Weaknesses:**

1) Comparative scope is limited. The experiments omit strong modern baselines, particularly Vision-Language Models (VLMs) such as CLIP, which are now standard in continual learning research, such as PET [1]?

2) The paper primarily benchmarks on traditional image classification datasets, missing the opportunity to demonstrate robustness in multi modal or open-vocabulary settings where pre-trained models excel.

3) Equation 2 supposed to be \mathcal{D}_t) ?

4) What is the advantages of this approach or how did the SAIM is aiding to mitigate forgetting. Though it performs better is several vision tasks without being compared with recent PTM based model it is hard to get the efficacy of the approach.

5) I can see EWC and ER, the methods before 2020 and ER is using replay. Does this method also use replay? If yes, the details on size of memory buffer is needed and the minimization equation (2) need to be updated.

6) Referring to Table 9, For example CIFAR100, is not 69.9 too low as compared to other VLM backed CL such as CLAP4CLIP [2] (~86).


References

[1] Wang, Liyuan, et al. "HIDE-PET: continual learning via hierarchical decomposition of parameter-efficient tuning." IEEE Transactions on Pattern Analysis and Machine Intelligence (2025).

[2] Jha, Saurav, Dong Gong, and Lina Yao. "Clap4clip: Continual learning with probabilistic finetuning for vision-language models." Advances in neural information processing systems 37 (2024): 129146-129186.

**Questions:**

Please check the weakness part

---

### Official Review · Reviewer_f2U8 · 2025-10-28

**Soundness:** 2
**Presentation:** 3
**Contribution:** 3
**Rating:** 4
**Confidence:** 4

**Summary:**

This manuscript considered the problem of continual learning with large pre-trained models. The authors proposed the Sharpness-Aware Isotropic Merging (SAIM) framework and introduced sharpness-aware optimizations in both the fine-tuning and model merging stages. Specifically, this manuscript introduced a Sharpness-Aware Block Coordinate Descent (SA-BCD) optimizer to guide the model towards the flatter minima and selectively update the most task-sensitive parameters to enhance the robustness and generalization ability. Furthermore, an adaptive isotropic merging algorithm was introduced to dynamically balance the singular value spectrum across tasks to avoid overemphasizing any single task and maintain balanced knowledge. Experiments on both vision and language tasks were conducted to demonstrate the superiority of the proposed SAIM framework.

**Strengths:**

1. This manuscript considers sharpness-aware optimization, which was underexplored in the context of continual learning.
2. The perspective of sharpness connected two stages of continual learning, continual fine-tuning and model merging, providing a unified view to bridge the gap between these two stages.
3. The proposed method was evaluated on both vision and language tasks with different models (i.e., ViT and LLM), showing its flexibility across different tasks and models.

**Weaknesses:**

1. One of my main concerns regarding the proposed method is the complexity and computational overhead compared to the existing methods. The proposed SAIM involved gradient manipulation and singular value decomposition in the two stages, which can introduce heavier computations in general. However, it seems that the authors didn't provide enough discussion regarding this aspect, even in the appendix.
2. The baselines in this manuscript only cover limited representative methods within the area of continual learning. More diverse baselines can be involved. Some methods, like the orthogonal-projection-based method, can be compared and discussed with the proposed method, especially considering that both of them are based on the gradient-constrained optimization.

**Questions:**

1. During training, how many copies of model parameters do we need to maintain? From Section 3.2, it seems we need to maintain $\theta_{t-1}$ and $\theta_{T_{t}}$ (two copies of a single model size). With Eq. (11), we also need to construct a new parameters via constructing a new diagonal. Furthermore, Eq. (12) indicated that we also need to maintain the initial model $\theta_{0}$. Thus, I wonder how many copies of model parameters should we keep during training? Could the authors provide any quantitative statistics to help the readers have a straightforward understanding about the training memory rquirement.
2. In Section 3.1, the authors claimed that "the magnitude of $|m_{t}|$ is used to select the top $p\%$ parameters with the largest absolute values $\Omega_{t}$". However, based on my experience, doing such a top-$p\%$ selection is usually time-consuming due to a sorting process. I wonder how this process affects the overall time complexity.
3. I checked the main body and the appendix and didn't find any discussion regarding the complexity discussion regarding the time, computation and memory. I recommend the authors provide more empirical comparisons to demonstrate the proposed method did not sacrifice the efficiency too much.
4. I noticed that the adopted baseline methods in the experimental part mainly cover the merging-based methods. It would be better if the authors could involve more comparisons with other orthogonal-projection-based or optimization-based methods that also considered the constrained optimization. The authors can refer to O-LoRA. InfLoRA, BiLoRA as examples.

References
- Orthogonal Subspace Learning for Language Model Continual Learning. EMNLP 2023.
- InfLoRA: Interference-Free Low-Rank Adaptation for Continual Learning. CVPR 2024.
- BiLoRA: almost-orthogonal parameter spaces for continual learning. CVPR 2025.

---

### Official Review · Reviewer_5A2E · 2025-10-29

**Soundness:** 2
**Presentation:** 3
**Contribution:** 2
**Rating:** 4
**Confidence:** 4

**Summary:**

This paper proposes Sharpness-Aware Isotropic Merging (SAIM), a joint fine-tuning and merging optimization framework for continual learning. Specifically, it introduces Sharpness-Aware Block Coordinate Descent (SA-BCD) to optimize toward flatter minima and alleviate parameter conflicts, alongside an adaptive isotropic merging mechanism to balance the knowledge across different tasks. Extensive experiments on both vision and language tasks validate the effectiveness of the proposed method.

**Strengths:**

1. The motivation behind the proposed method is clear. Sharpness-aware minimization is a well-established and effective technique in continual learning, and its application within a merging-based framework is intuitive.
2. The paper presents a comprehensive and well-structured study. The proposed approach is supported by convergence analysis, and the authors provide extensive experimental results and discussion in main paper and appendix, covering three vision backbones and two LLMs evaluated on multiple benchmarks.

**Weaknesses:**

1. There are several prior works in continual learning have explored similar ideas of model merging (also referred to as interpolation or ensembling). However, these studies appear to be largely omitted from both experimental comparisons and related work discussion. The authors are encouraged to incorporate these relevant works for a fairer and more complete evaluation.

[1]  Stojanovski, Z., Roth, K., and Akata, Z. Momentum-based weight interpolation of strong zero-shot models for con tinual learning. In NeurIPS Workshop, 2022.

[2]  Simon, C., Faraki, M., Tsai, Y.-H., Yu, X., Schulter, S., Suh, Y., Harandi, M., and Chandraker, M. On generalizing beyond domains in cross-domain continual learning. In CVPR, 2022.

[3]  Lee, J., Joo, D., Hong, H. G., and Kim, J. Residual continual learning. In AAAI, 2020.

[4]  Lin, G., Chu, H., and Lai, H. Towards better plasticity stability trade-off in incremental learning: A simple linear connector. In CVPR, 2022.

[5]  Marouf, I. E., Roy, S., Tartaglione, E., and Lathuili`ere, S. Weighted ensemble models are strong continual learners. In ECCV, 2024.

[6] Li, M., Lu, Y., Dai, Q., Huang, S., Ding, Y., Lu, H., BECAME: Bayesian Continual Learning with Adaptive Model Merging. In ICML, 2025.

2. Given that this work targets continual learning, it would be more convincing to include comparisons with recent state-of-the-art methods, rather than limiting the scope to merging-based baselines. It is worth noting that EWC and ER were proposed over five years ago, and thus may not represent the current frontier of the field.
3. Another concern is the lack of computational efficiency analysis. It is important to report the computational and memory costs associated with the fine-tuning and merging processes in SAIM. A comparison with prior continual learning methods on these aspects would strengthen the empirical evaluation.

**Questions:**

1. Why is the BWT metric not reported for the LLM benchmarks? BWT is a key measure of the stability–plasticity trade-off in continual learning and should be included for consistency with the vision benchmarks.
2. In Section 3.2 L259, the authors state that "\alpha is a scaling factor determined by validation set search". To clarify, are all benchmarks and datasets divided into training, validation, and test subsets? If not, there may be concerns regarding potential data leakage. Similarly, it would be helpful to summarize the treatment of other hyperparameters, such as perturbation magnitude, parameter selection ratio, and balance coefficient, detailing how they are determined and whether the proposed method is robust across different combinations of these factors.

---

### Official Review · Reviewer_k1Co · 2025-11-01

**Soundness:** 3
**Presentation:** 2
**Contribution:** 2
**Rating:** 2
**Confidence:** 3

**Summary:**

This paper SA-BCD, a framework which jointly optimizes fine-tuning processes and subsequent merging stages to improve continual learning performance. Empirical results demonstrate good performance across vision and NLP tasks.

**Strengths:**

1. The paper is overall well-written and easy to follow.
2. The results demonstrate good empirical performance across vision and NLP tasks, including results on LLMs (Llama).

**Weaknesses:**

1. The proposed method incurs high computational and storage cost in several aspects.

a. Using SAM during fine-tuning doubles the computational cost for gradient computation.

b. Storing the gradient momentum (for masking gradient during fine-tuning) leads to extra storage cost during fine-tuning.

2. SA-BCD offers modest novelty in methodological contributions, as it draws ideas from previous literature in both fine-tuning stage and merging stage.

a. During the fine-tuning stage, the proposed method utilizes SAM [1].

b. During the merging stage, it draws the idea of isotropic model merging from Iso-C [2], and extends to a continual learning scenario.

3. SA-BCD breaks several key requirements as a continual model merging method, due to the following two reasons.

a. While the authors criticize previous model merging methods for “the fine-tuning and merging processes are often treated as independent stages”, this independence is an essential property of model merging paradigm. Only when the merging process is decoupled from fine-tuning, it can be general enough to utilize the publicly available fine-tuned checkpoints without access to the training data for fine-tuning (i.e., you can not expect the available checkpoints were fine-tuned the way you expected).

For example, the baseline model merging methods (such as task arithmetic, DARE and TIES) can utilize the fine-tuned Llama-3 models directly provided by the Mergebench and the merging only takes minutes. In contrast, the proposed framework needs to fine-tune a Llama-3 model respectively on these tasks using its custom fine-tuning strategy before merging, thereby defeating one key advantage of model merging.

b. The fine-tuning starts from the merged model with the (t-1) tasks rather than the pre-trained model, leading to another constraint that the method continuously requires training data for all tasks.

Therefore, the proposed method departs from the definition as a continual model merging method and should be presented as a continual learning category that incorporates techniques from model merging domain. As a result, the performance comparison to the model merging method (e.g. Figure 1) seems unfair, and it should be compared to the state-of-the-art continual learning methods instead.

[1] Sharpness-Aware Minimization for Efficiently Improving Generalization
[2] No Task Left Behind: Isotropic Model Merging with Common and Task-Specific Subspaces

**Questions:**

See weaknesses

---

### Meta-Review · Area_Chair_EkMY · 2025-12-09

**Summary:**

This paper presents the Sharpness-Aware Isotropic Merging (SAIM) framework to enhance continual learning with large pre-trained models by addressing issues related to catastrophic forgetting and interference. The proposed approach introduces a Sharpness-Aware Block Coordinate Descent (SA-BCD) optimizer and adaptive isotropic merging to improve convergence and robustness.

While reviewers appreciate the clarity of the manuscript and motivations behind the proposed approach, they noted a high computational and storage cost associated with the proposed methods, as well as concerns regarding novelty, since SAIM draws from existing techniques without sufficient new insight. Additionally, the discussions around computational efficiency and scalability appear inadequate, particularly given the lack of discussion of and comparison with other CL methods based on in the literature. The authors did not provide a rebuttal, and the recommendation is to Reject.

**Reviewer Concerns:**

Authors provided no rebuttal.

**Reviewer Scores:**

Authors provided no rebuttal, I don't believe the reviewers would have been convinced in any case.

---

### Decision · Program_Chairs · 2026-01-26

Reject